# Simple Approximation and Derivative Free Inference-Time Scaling for Diffusion Models via Sequential Monte Carlo on Path Measures

Chenyang Wang [* 1]   Weizhong Wang [* 2]   Yinuo Ren [3]   Jose Blanchet [3 4]   Yiping Lu [5]

## Abstract

Diffusion-based generative models increasingly rely on inference-time guidance, adding a drift term or reweighting mixture of experts, to improve sample quality on task-specific objectives. However, most existing techniques require repeated score or gradient evaluations, introducing bias, high computational overhead, or both. We introduce URGE, approximation-free Resampling via Girsanov Estimation, a derivative-free inference-time scaling algorithm that performs pathwise importance reweighting via a Girsanov change of measure. Instead of computing gradient-based particle weights in previous work, URGE attaches a simple multiplicative weight to each simulated trajectory and periodically resamples. No score, no Hessian, and no PDE evaluation is required. We establish an equivalence between pathwise and particle-wise SMC: the Girsanov path weight admits a backward conditional expectation that recovers the previous particle-level weights, guaranteeing that both schemes produce the same approximation-free terminal law. Empirically, URGE outperforms existing inference-time guidance baselines on synthetic tests and diffusion-model benchmarks, achieving better generation quality, while being significantly simpler to implement and fully gradient-free.

*Equal contribution [1]School of Mathematical Sciences, Peking University, Beijing, China [2]School of Mathematical Sciences, Fudan University, Shanghai, China [3]Institute for Computational & Mathematical Engineering, Stanford University, Stanford, CA, United States [4]Management Science & Engineering, Stanford University, Stanford, CA, United States [5]Department of Industrial Engineering & Management Sciences, Northwestern University, Evanston, IL, United States. Correspondence to: Yiping Lu <yiping.lu@northwestern.edu>.

*Proceedings of the 43rd International Conference on Machine Learning*, Seoul, South Korea. PMLR 306, 2026. Copyright 2026 by the author(s).

## 1. Introduction

Modern generative models have emerged as a powerful paradigm for learning complex, high-dimensional data distributions. In particular, diffusion models (Ho et al., 2020; Sohl-Dickstein et al., 2015; Song and Ermon, 2019; Song et al., 2020) and flow-based methods (Zhang et al., 2018a; Lipman et al., 2022; Albergo and Vanden-Eijnden, 2022; Liu et al., 2022) provide a principled and scalable framework for generative modeling, achieving state-of-the-art performance across diverse applications, including video generation (Ho et al., 2022), protein design (Gruver et al., 2023), and large-scale text generation (Li et al., 2022; Nie et al., 2025). A unifying perspective underlying these approaches is their formulation in terms of stochastic differential equations (SDEs) (Song et al., 2020; Lipman et al., 2022; Albergo et al., 2025; Liu et al., 2022). Concretely, generation can be viewed as simulating a carefully designed SDE

$$\mathrm{d}X_t = v(X_t, t)\mathrm{d}t + V(t)\mathrm{d}W_t, \quad X_0 \sim \mathcal{N}(0, I_d)$$

with an appropriate learned drift field $v : \mathbb{R}^d \times \mathbb{R}_{\geq 0} \to \mathbb{R}^d$ and time-dependent diffusion $V : \mathbb{R}_{\geq 0} \to \mathbb{R}$ coefficient transform a simple base distribution into the data distribution. $W_t$ denotes a standard $d$-dimensional Brownian motion. Forward simulation of this SDE yields samples whose distribution approximates the target $p_{\mathrm{data}}$.

While such models are typically trained to faithfully capture the data distribution $p_{\mathrm{data}}$, deployment-time requirements often extend beyond unconditional sampling. Users seek to improve generation quality or enforce downstream objectives, *e.g.*, physical validity or image-text alignment, without retraining the model. This motivates *inference-time scaling*, which aims to steer generation at inference while reusing a pretrained model. Formally, task-oriented objectives can be expressed through a reward or preference function $\mathbf{r}(x)$ and inference-time scaling amounts to sampling from a target distribution that incorporates both data and task objectives: $q(x) \propto p_{\mathrm{data}}(x)\mathbf{r}(x)$ (Uehara et al., 2025).

A common approach to incorporating such a reward during inference is guidance, which directly alters the generative stochastic dynamics to bias samples toward high-reward regions (Hong, 2024; Wu et al., 2023a; Yoon et al., 2024; Castillo et al., 2023; Dhariwal and Nichol, 2021; Ho and

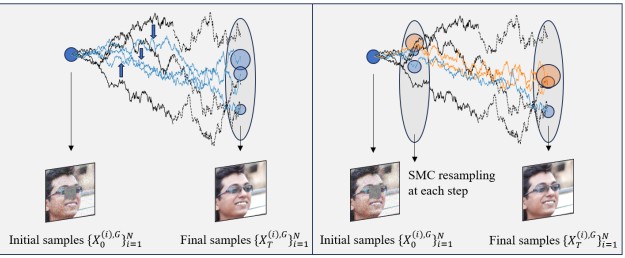

(a) Naïve Guidance    (b) Unbiased Guidance via SMC

*Figure 1.* Step-wise resampling corrects suboptimal guided generation toward the reward-tilted distribution. (a) In Naive Guidance, blue trajectories follow the guided diffusion $\mathrm{d}X_t^G = \big(v(X_t^G, t) + V(t)^2 \nabla_x G(X_t^G, t)\big)\mathrm{d}t + V(t)\mathrm{d}W_t$, whereas black trajectories follow the base diffusion without $G$. (b) In our algorithm, we reweight and resample the generation trajectory to obtain consistent posterior samples from the reward-tilted distribution.

Salimans, 2022; Sadat et al., 2024; Nichol et al., 2021; Rombach et al., 2022; Saharia et al., 2022; Podell et al., 2023; Gruver et al., 2023). In the SDE formulation, guidance augments the drift of the sampling process with an additional gradient term derived from a guidance potential $G : \mathbb{R}^d \to \mathbb{R}$, leading to the guided SDE

$$dX_t^G = (v(X_t^G, t) + V(t)^2 \nabla_x G(X_t^G, t))\mathrm{d}t + V(t)\mathrm{d}W_t,$$

where the additional term $V(t)^2 \nabla G$ steers generative trajectories toward regions favored by the downstream objective.

Despite its empirical effectiveness, guidance leaves open a fundamental question: *what distribution is the guided model actually sampling from?* From a mathematical standpoint, exact sampling from the tilted distribution $q(x) \propto p_{\text{data}}(x)\mathbf{r}(x)$ via a modified diffusion process requires the Doob's $h$-transform $h(x, t) = \mathbb{E}\left[\mathbf{r}(X_T) \mid X_t = x\right]$ (Denker et al., 2024; Tang and Xu, 2024; Nguyen et al., 2025; Sabour et al., 2025; Li et al., 2024; Yang et al., 2024; Zhu et al., 2026) of the original generative dynamics, which represents the expected future reward attainable from an intermediate state under the original dynamics. In this construction, the optimal guidance should be adjusted by $\nabla_x \log h(x, t)$, therefore steering particles toward states favorable with respect to the expected final reward. However, constructing the correct $h$-function typically requires solving an associated backward Kolmogorov equation, which often requires extra training in high-dimensional generative models (Domingo-Enrich et al., 2024; Havens et al., 2025; Liu et al., 2025; Albergo and Vanden-Eijnden, 2024). Consequently, practical guidance methods resort to inherently approximated strategies, applying local drift modifications while partially ignoring future trajectories. A natural question arises:

> **Can we design an approximation-free inference-time scaling procedure, even with suboptimal guidance?**

To address this challenge, we introduce **URGE** (**U**nbiased **R**esampling via **G**irsanov **E**stimation), an approximation-free resampling framework that eliminates the bias induced by suboptimal guidance. Recent works (Chen et al., 2025; Ren et al., 2025a; Wu et al., 2023a; Skreta et al., 2025; Lee et al., 2025; He et al., 2025; Zhu and Lu, 2026; Wei et al., 2026) have explored inference-time scaling by correcting defects in the generation probability (Fan et al., 2025) through reweighting generated samples, most commonly by integrating diffusion models with Sequential Monte Carlo (SMC) methods (Martino et al., 2017; Del Moral et al., 2006; 2012; Doucet et al., 2000) methods to obtain approximation-free, approximation-free estimators of the target distribution by resampling from diverse trajectories from an underlying SDE. However, these approaches often rely on the infinitesimal generators of the SDEs and thus require access to higher-order derivatives of the reward function. Such requirements incur substantial computational overhead and may limit practical applicability to large-scale reward functions. **In contrast, rather than reweighting data after their being generated, URGE resamples the generation process itself by applying SMC directly to the generation trajectory (path measure) of the underlying diffusion.** This leads to a mixture-of-expert type inference-time scaling: simulate multiple trajectories under $\mathbb{P}^G$, reweight them using the path-space likelihood ratio, and resample. Even when the guidance $G$ is imperfect, this correction removes its bias, ensuring convergence of the empirical terminal distribution to the true posterior $q$.

Theoretically, we reveal a fundamental marginalized equivalence between resampling in path space (URGE) and particle space (FK-Corrector/AFDPS), demonstrating that lifting resampling from particle space to path space enables substantially more flexible weight construction and yields an easily implementable, fully derivative-free weighting scheme. Empirically, we validate the effectiveness of our inference-time scaling strategy on both synthetic datasets and real-world applications, showing that URGE achieves strong performance comparing with state-of-the-art baselines, while requiring no derivative information from the likelihood.

**Contribution** Building on this insight, our work makes the following contributions:

- We propose a novel mixture-of-experts inference-time scaling method that integrates SMC with generative processes. Our method performs resampling over generation trajectories rather than over generated samples. Our resampling weights are computed via a Girsanov-type change of measure with respect to the filtration of each diffusion path that does not require reward derivatives, extending its applicability to neural reward models, such as those used in prompt to image generation.

- We identify the equivalence that, conditional on the terminal values, the limit of the pathwise reweighting term recovers the infinitesimal reweighting of previous approaches (FK-Corrector/AFDPS), while URGE lifts the construction of resampling weights from the particle space to the path space and enables more flexible and accurate weight design and evaluation.
- Empirically, we validate the effectiveness of our inference-time scaling strategy: URGE consistently outperforms state-of-art baselines on various tasks across synthetic and real-world tasks, while utilizing no derivative information of the likelihood.

## 2. Methodology: approximation-free Guidance via Girsanov Sequential Monte Carlo

We consider the probability mass on the measurable space $(\Omega_T, \mathcal{F}_T)$, where $\Omega_T = C_0([0,T], \mathbb{R}^d)$ and $\mathcal{F}_T$ denotes the $\sigma$-algebra generated by the generative process $X_{0:T}$, which we take as reference and obeys the following SDE:

$$\mathrm{d}X_t = v(X_t, t)\mathrm{d}t + V(t)\mathrm{d}W_t, \quad X_0 \sim \mathcal{N}(0, I_d),$$

where $v : \mathbb{R}^d \times [0,T] \to \mathbb{R}^d$ is a drift field, $V : [0,T] \to (0,\infty)$ is a time-dependent diffusion schedule, and $W_t$ is a standard $d$-dimensional Brownian motion under the probability measure $\mathbb{P}$. We denote $p(x,t)$ as the marginal distribution at time $t$, *i.e.*, the law of $X_t$ under $\mathbb{P}$, and $p_{\text{data}} = p_T$ as the final generated distribution.

### 2.1. Inference-Time Reward-Tilting

At inference time, requirements often extend beyond unconditional sampling. To accommodate such requirements, one typically specifies a task-dependent reward or reweighting function $r : \mathbb{R}^d \times [0,T] \to \mathbb{R}$ that encodes the desired inference objectives, satisfying $r(x,T) \equiv \mathbf{r}(x)$. This induces a distribution path:

$$q(x,t) \propto p(x,t)e^{r(x,t)}, \tag{1}$$

with the target terminal distribution being the reward-tilted posterior $q(x,T) \propto p_{\text{data}}(x)e^{\mathbf{r}(x)}$. The function $r(x,t)$ serves as a time-dependent interpolation between the initial reward $r(x,0)$, usually chosen as 0, and the terminal reward $\mathbf{r}(x)$, gradually incorporating the reward signal into tilting. In inverse problems, a convenient choice is to set $r(x,t) \equiv \mathbf{r}(x)$ for all $t$. In contrast, for prompt engineering, a common choice is $r(x,t) = g(t)\mathbf{r}(x)$, where $g(t)$ is a scalar interpolation with $g(0) = 0$ and $g(T) = 1$.

However, since this posterior is up to an unknown normalization constant and typically defined implicitly via $p_{\text{data}}$ and $\mathbf{r}$, direct sampling from $q$ is in general intractable. A simple yet effective method is guidance $X^G = \{X_t^G\}_{t\in[0,T]}$

evolves according to

$$\mathrm{d}X_t^G = \big(v(X_t^G, t) + V(t)^2\nabla_x G(X_t^G, t)\big)\mathrm{d}t + V(t)\mathrm{d}W_t, \tag{2}$$

where $X_0^G \sim \mathcal{N}(0, I_d)$ and $G : \mathbb{R}^d \times [0,T] \to \mathbb{R}$ is a guidance potential, which encodes inference-time preferences, such as classifier, alignment scores, or surrogate rewards. Theoretically speaking, incorporating this drift modification induces a change of measure over sample paths and consequently alters the path measure of generated trajectories (*cf.* (4)). However, **the altered path measure induced by guidance does not necessarily coincide with the desired reward-tilted path measure** (Chidambaram et al., 2024).

### 2.2. Infinitesimal Perspective: Particle-Space SMC

To address this challenge, previous works (FK Corrector (Skreta et al., 2025)/ AFDPS (Chen et al., 2025)) propose particle-space SMC correctors that can be understood as an correction to the infinitestimal generator via reweighting instances after being generated. Let $\mathcal{L}_t^G$ denote the generator of the guided diffusion (2) acting on smooth test function $\varphi \in C_b^2(\mathbb{R}^d)$:

$$\mathcal{L}_t^G\varphi(x) = \frac{1}{2}\mathrm{tr}\big(V^2(t)\nabla^2\varphi(x)\big) + \big(v(x,t) + V^2(t)\nabla_x G(x,t)\big)^\top\nabla\varphi(x).$$

The particle-space SMC idea is to augment propagation by an additional state-dependent potential $w_{\text{AFDPS}}(\cdot, t)$ so that the resulting law evolves toward the reward-tilted marginals.

**Theorem 2.1.** *The reward-tilted distribution path corresponds to the following generator:*

$$\mathcal{L}_t^{\text{eff}} = \mathcal{L}_t^G + w_{\text{AFDPS}}(\cdot, t), \tag{3}$$

*where the potential is given by:*

$$w_{\text{AFDPS}}(x,t) = \Big(\partial_t r - \tfrac{1}{2}V^2(t)\big(\Delta_x r + \|\nabla_x r\|_2^2\big)$$
$$+ \nabla_x r^\top\big(v + V^2(t)\nabla_x G\big) + V^2(t)\Delta_x G$$
$$+ V^2(t)\nabla_x\log p_t(x)^\top\nabla_x(G - r)\Big)(x,t).$$

We refer readers to Appendix B.1 for more details.

### 2.3. Our Path Perspective: URGE via Path Weights

Rather than debiasing the guidance by reweighting individuallu generated instants, we propose to operate on the generation trajectories, *i.e.*, reweighting and resampling over path space. This lifting provides additional flexibility in constructing importance weights.

Let $\mathbb{P}^G$ denote the path measure induced by the guided SDE, defined on the measurable space $(\Omega_T, \mathcal{F}_T)$. For any time

| Method | Reweighting |
|---|---|
| FK Steering[1] (Singhal et al., 2025) | $r(X_{t+\Delta t}, t + \Delta t) - r(X_t, t)$ |
| FK Corrector (Skreta et al., 2025) AFDPS (Chen et al., 2025) | $\int_t^{t+\Delta t} \left( \partial_\tau r - \dfrac{V^2(t)}{2}(\Delta_x r + \|\nabla_x r\|_2^2) + \nabla_x r^\top (v + V^2(t)\nabla_x G) \right.$ $\left. + V^2(t)\nabla_x \log p_t^\top \nabla_x (G - r) + V^2(t)\Delta_x G \right)(X_\tau^G, \tau)\mathrm{d}\tau$ |
| URGE (Ours) | $\underbrace{\int_t^{t+\Delta t} -V(\tau)\nabla_x G(X_\tau^G, \tau)^\top \mathrm{d}W_\tau - \dfrac{1}{2}\int_t^{t+\Delta t} V^2(\tau)\|\nabla_x G(X_\tau^G, \tau)\|_2^2\mathrm{d}\tau + \int_t^{t+\Delta t} \mathrm{d}r(X_\tau^G, \tau)}_{\text{Path Information}}$ |

*Table 1.* Comparison of reweighting functions induced by different steering methods. Unlike prior approaches, URGE uniquely constructs reweighting functions with respect to the full generative process.

$t$, by Girsanov's theorem, the Radon-Nikodym derivative between the uncontrolled measure $\mathbb{P}$ and the guided measure $\mathbb{P}^G$ is

$$\frac{\mathrm{d}\mathbb{P}}{\mathrm{d}\mathbb{P}^G}(X_{0:t}^G) \propto \exp\left( - \int_0^t V(s)\nabla_x G(X_s^G, s)^\top \mathrm{d}W_s \right.$$
$$\left. - \frac{1}{2}\int_0^t V^2(s)\|\nabla_x G(X_s^G, s)\|_2^2 \mathrm{d}s \right). \quad (4)$$

Then we define a new path measure $\mathbb{Q}$ on the space of continuous trajectories, via the Radon-Nikodym derivative

$$\frac{\mathrm{d}\mathbb{Q}}{\mathrm{d}\mathbb{P}}(X_{0:t}) \propto \exp\left( r(X_t, t) - r(X_0, 0) \right).$$

Under this construction, the terminal marginal of $\mathbb{Q}$ exactly matches the reward-tilted posterior $q(x, t) \propto p(x, t)e^{\mathbf{r}(x)}$, when evolving from the initial distribution $q(x, 0) \propto p(x, 0)e^{r(x,0)}$.

The discrepancy between the reward-tilted distribution and the guided trajectory distribution can be characterized using the chain rule for Radon-Nikodym derivatives. For a generation trajectory $X_{0:T}^G$, the importance weight that corrects samples drawn from the guided generation trajectory distribution $\mathbb{P}^G$ toward the target distribution $\mathbb{Q}$ is

$$\frac{\mathrm{d}\mathbb{Q}}{\mathrm{d}\mathbb{P}^G}(X_{0:t}^G) = \frac{\mathrm{d}\mathbb{Q}}{\mathrm{d}\mathbb{P}}(X_{0:t}^G)\frac{\mathrm{d}\mathbb{P}}{\mathrm{d}\mathbb{P}^G}(X_{0:t}^G)$$
$$\propto \exp\left( r(X_t^G, t) - r(X_0^G, 0) - \int_0^t V(s)\nabla_x G(X_s^G, s)^\top \mathrm{d}W_s \right.$$
$$\left. - \frac{1}{2}\int_0^t V^2(s)\|\nabla_x G(X_s^G, s)\|_2^2 \mathrm{d}s \right). \quad (5)$$

We apply an SMC estimator that resamples generation trajectories in proportion to the importance weight $\frac{\mathrm{d}\mathbb{Q}}{\mathrm{d}\mathbb{P}^G}$, *i.e.*, rather than generating a single instance, we generate multiple trajectories and select and resample higher-quality generations according to their importance weights.

[1]As per the official implementation of FK-Steering (Singhal et al., 2025): https://github.com/zacharyhorvitz/Fk-Diffusion-Steering/

### 2.4. URGE Implementation

Although our resampling strategy could perform resampling after an arbitrarily long time horizon, in practice we apply SMC resampling at every step to achieve better performance. Generally speaking, the pathwise weight associated with URGE is defined by the following function, where $X_{s:t}^G$ denotes the trajectory of particle $X^G$ on an arbitrary time interval $[s, t]$:

$$w_{s,t}^{\text{URGE}}(X_{s:t}^G) = \exp\left( \int_s^t -V(\tau)\nabla_x G(X_\tau^G, \tau)^\top \mathrm{d}W_\tau \right.$$
$$\left. - \frac{1}{2}\int_s^t V^2(\tau)\|\nabla_x G(X_\tau^G, \tau)\|_2^2\mathrm{d}\tau + \int_s^t \mathrm{d}r(X_\tau^G, \tau) \right), \quad (6)$$

where $\mathrm{d}r(X_\tau^G, \tau)$ denotes the Itô differential of the reward.

Specifically, during inference time, we discretize the time interval $[0, T]$ into a schedule $\{t_k\}_{k=0}^K$, where $t_k = k\Delta t$ and $\Delta t_k = t_{k+1} - t_k$. On each interval $[t_k, t_{k+1}]$, we simulate $N$ trajectories $\{X_t^{(i),G}\}_{i=1}^N$ following the guided SDE (2), via a single update. It is crucial to use sufficiently small time step $\Delta t_k$ to ensure stable importance weighting. With small time steps, the incremental change in the state $X_t^{(i),G}$ remains limited, and the associated Radon-Nikodym derivative varies more smoothly across particles, since the reward $r$ exhibits only minor variation. This consequently reduces the variance of the resampling procedure.

At each step, particles are resampled according to their importance weights, yielding an approximation of the target trajectory $\mathbb{Q}$. The simplest way to compute the weight of path $X_t^{(i),G}$ on a small time interval $[s, t]$ is given by

$$\beta_{s,t}^{(i)} = \exp\left( r(X_t^{(i),G}, t) - r(X_s^{(i),G}, s) \right.$$
$$- V(s)\nabla_x G(X_s^{(i),G}, s)^\top \sqrt{t - s}\,\xi_s^{(i)} \quad (7)$$
$$\left. - \frac{1}{2}V^2(s)\|\nabla_x G(X_s^{(i),G}, s)\|_2^2 (t - s) \right),$$

which matches the Euler-Maruyama (EM) discretization of the likelihood ratio above (5). After computing $\beta_{t,t+\Delta t}^{(i)}$,

the proposed states $X_{s,t}^{(i),G}$ are resampled with probabilities proportional to $\beta_{s,t}^{(i)}$. From the Feynman-Kac perspective, such resampling implements a birth-death mechanism, in which large-weight particles are replicated while low-weight ones are eliminated (Del Moral, 2004; Del Moral and Miclo, 2000). The overall algorithm is summarized in Algorithm 1.

We summarize different importance weight constructions in Table 1. Compared to `FK Steering` (Singhal et al., 2025), our method incorporates additional pathwise informative terms, which ensures that the resulting resampling procedure is approximation-free. Moreover, compare to `FK Corrector` (Skreta et al., 2025) and `AFDPS` (Chen et al., 2025) that require access to the score function or the Laplacian of $r$, our formulation avoids such quantities entirely. This property makes our approach particularly suitable for practical scenarios with black-box reward functions, where such derivatives are unavailable or expensive to compute. Also by incorporating the Itô integral in the weight, URGE encapsulates stochastic path information into resampling, in contrast to being full deterministic in previous methods, leading to better performance. The most important advantage of URGE is that **URGE lifts importance weighting from generated samples to full diffusion paths, which provides greater flexibility in designing importance weights.** In principle, one may adopt any numerical scheme to the path weight (6), instead of the EM scheme used in (7) for more accurate approximations. For example, when computational cost is limited, one may adopt sparser discretization schemes complemented with higher-order schemes for comparable performance.

## 3. Theoretical Analysis: Equivalence between Path and Particle Spaces

In this section, we establish the equivalence between the pathwise reweighting induced by URGE and the particle-level reweighting employed by `FK-Corrector` (Skreta et al., 2025) and `AFDPS` (Chen et al., 2025) at the level of generators that characterize the infinitesimal evolution of the generation Markov process. We show that marginalizing the URGE path-space importance weights yields an effective particle-level reweighting, which exactly recovers the weighting structure used by `FK-Corrector` and `AFDPS`, confirming the correctness of our method.

We first establish the approximation-freeness of URGE. Specifically, we show that reweighting step preserves the mean of weighted empirical measure (See Appendix A.2 for further discussion). Let $\mathbb{P}$ be the law of the base diffusion with generator $L$, and let $\overline{\mathbb{P}}$ denote the law of URGE. For $\varphi \in C_b^{2,1}(\mathbb{R}^d \times \mathbb{R})$ and $t_k \leq t < t_{k+1}(0 \leq k \leq K-1)$, define $M_t^N(\varphi) := \sum_{i=1}^N \frac{\beta_{t_k,t}^{(i)}}{\sum_{j=1}^N \beta_{t_k,t}^{(j)}} \varphi(X_t^{(i),G}, t)$.

---

**Theorem 3.1** (Continuous approximation-freeness). *For any test function $\varphi \in C_b^{2,1}(\mathbb{R}^d \times \mathbb{R})$, if the distribution at time $s$ satisfies $\mathbb{E}_{\overline{\mathbb{P}}}[M_s^N(\varphi)] = \mathbb{E}_{\mathbb{P}}[\varphi(X_s, s)]$, then when $\Delta t \to 0$ and $N \to \infty$, for any time $t$, we have*

$$\mathbb{E}_{\overline{\mathbb{P}}}[M_t^N(\varphi)] = \frac{\mathbb{E}_{\mathbb{P}}[e^{r(X_t,t)-r(X_s,s)}\varphi(X_t,t)]}{\mathbb{E}_{\mathbb{P}}[e^{r(X_t,t)-r(X_s,s)}]}.$$

*Specifically, by taking $s = 0$, at terminal time $T$,*

$$\mathbb{E}_{\overline{\mathbb{P}}}[M_T^N(\varphi)] = \frac{\mathbb{E}_{\mathbb{P}}[e^{\mathbf{r}(X_T)-r(X_0,0)}\varphi(X_T,T)]}{\mathbb{E}_{\mathbb{P}}[e^{r(X_T)-r(X_0,0)}]}.$$

The proof is given in Appendix B.2. This theorem ensures that marginalizing the URGE path measure does not introduce systematic distortion in the terminal-time distribution.

Next, we analyze the terminal-time marginal of URGE pathwise measure and show that it induces the same particle-level reweighting as `FK-Corrector` and `AFDPS`. We first study the marginalized generator of URGE via exploiting the Feynman–Kac duality by analyzing the backward value function on the $i$-th particle:

$$\psi(x,t) := \mathbb{E}\left[M_t^N(\varphi) \mid X_t^{(i),G} = x\right].$$

This formulation allows us to identify the effective marginalized particle space generator of the reweighted process via the associated Kolmogorov backward equation.

---

**Lemma 3.2** (Effective Marginalized Generator). *Define infinitesimal generator $\mathcal{L}_t^G$ of the process $(X_t^G)_{t \in [0,T]}$ for $\varphi \in C_b^{2,1}(\mathbb{R}^d \times \mathbb{R})$. Assume that for each $(t,x)$, the instantaneous intensity of the weight exists and is given by*

$$\lambda(x,t) := \lim_{h \downarrow 0} \frac{1}{h}\left(\mathbb{E}_{\mathbb{P}^G}\left[w_{t-h,t}^{\text{URGE}}(X_{t-h:t}^{(i),G})\middle| X_t^{(i),G} = x\right] - 1\right).$$

*Then as $N \to \infty$, the value function $\psi(x,t)$ satisfies the generalized Kolmogorov backward equation:*

$$\partial_t \psi(x,t) + \mathcal{L}_t^G \psi(x,t) + (\lambda(x,t) - \bar{\lambda}_t)\psi(x,t) = 0.$$

*where $\bar{\lambda}_t = \mathbb{E}_{\overline{\mathbb{P}}}[\lambda(X_s^G, s)]$ is a normalizing factor.*

---

Theorem 3.2 shows that the effective marginalized generator on the particle space induced by path-space reweighting takes the form $\mathcal{L}_t^{\text{eff}} := \mathcal{L}_t^G + \lambda(\cdot, t)$. This generator can be interpreted as the generator of a mixture-of-experts SMC estimator on the particle space, augmented by the potential $\lambda(\cdot, t)$, with normalization $\bar{\lambda}_t$ coincides with that in `AFDPS` (Chen et al., 2025). In other words, **path-space resampling is therefore equivalent to particle-space resampling with weights given by expectation of the path-space**

**Algorithm 1** URGE: **U**nbiased **R**esampling via **G**irsanov **E**stimation

---

**Require:** Number of particles $N$, time grid $\{t_i\}_{i=0}^K$ with $t_0 = 0, t_K = T$; initial particles $\{X_0^{(i),G}\}_{i=1}^N$ with $X_0^{(i),G} \sim \mathcal{N}(0, I_d)$; drift $v$; guidance potential $G$; variance schedule $V(t)$; reward function $r$.

1: **for** $k = 0$ to $K - 1$ **do**
2:     **for** $i = 1$ to $N$ **do**
3:         Simulate particle dynamics (via Euler-Maruyama), where $\xi_{t_k}^{(i)} \sim \mathcal{N}(0, I_d)$:

$$X_{t_{k+1}}^{(i),G} = X_{t_k}^{(i),G} + \left( v(X_{t_k}^{(i),G}, t_k) + V(t_k)^2 \nabla_x G(X_{t_k}^{(i),G}, t_k) \right)(t_{k+1} - t_k) + V(t_k)\sqrt{t_{k+1} - t_k}\xi_{t_k}^{(i)}.$$

4:         Weight update:
$$\beta_{t_k,t_{k+1}}^{(i)} = \exp\left( r(X_{t_{k+1}}^{(i),G}, t_{k+1}) - r(X_{t_k}^{(i),G}, t_k) - V(t_k)\nabla_x G(X_{t_k}^{(i),G}, t_k)^\top \sqrt{t_{k+1} - t_k}\xi_{t_k}^{(i)} \right.$$
$$\left. - \tfrac{1}{2}V(t_k)^2 \|\nabla_x G(X_{t_k}^{(i),G}, t_k)\|_2^2 (t_{k+1} - t_k) \right).$$

5:     **end for**
6:     Resampling: $\{X_{t_{k+1}}^{(i),G}\}_{i=1}^N \sim \text{Categorical}\left( \{X_{t_{k+1}}^{(i),G}\}_{i=1}^N, \left\{ \frac{\beta_{t_k,t_{k+1}}^{(i)}}{\sum_{j=1}^N \beta_{t_k,t_{k+1}}^{(j)}} \right\}_{i=1}^N \right).$
7: **end for**

---

**weight conditioned on the terminal position**.

We observe a structural similarity between the generators induced by our proposed URGE method (Lemma 3.2) and FK-Corrector/AFDPS (Theorem 2.1). In both cases, the original guided-diffusion generator is augmented with an additional reweighting term. To establish marginalized equivalence between URGE and FK-Corrector/AFDPS, it therefore suffices to verify that these reweighting terms coincide. We now derive the explicit form of $\lambda(\cdot, t)$ for URGE. The following theorem establishes that the instantaneous growth rate $\lambda(\cdot, t)$ coincides exactly with the particle reweighting function $w_{\text{AFDPS}}(\cdot, t)$ used in FK-Corrector/AFDPS.

> **Theorem 3.3** (Marginalized Equivalence between URGE and FK-Corrector/AFDPS). *Let $p_s \in C^2(\mathbb{R}^d)$ denote the time-marginal density of (2) at time $s$. The convergence rate holds:*
>
> $$\lim_{s \to t} \frac{\mathbb{E}_{\mathbb{P}^G}\left[ w_{s,t}^{URGE}(X_{s:t}^G) \big| X_t^G = x \right] - 1}{t - s} = w_{AFDPS}(x, t). \tag{8}$$

This result indicates that URGE and FK-Corrector/AFDPS are in fact simulating the same underlying generative process, with URGE **lifting the construction of resampling importance weights from the particle space to the path space. This reformulation enables more flexible weight construction and facilitates a derivative-free inference-time scaling strategy.**

# 4. Experiment

In this section, we evaluate the empirical performance of URGE on reward-tilting tasks: a synthetic Gaussian Mixture Model, text-to-image generation and inverse prob-

lems. We show that our derivative-free URGE method consistently outperforms the discrete-time steering baseline FK-steering (Singhal et al., 2025) across all tasks, and matches the performance of the higher-order, derivative-based FK-Corrector (Skreta et al., 2025)/AFDPS (Chen et al., 2025) when derivatives are available, demonstrating the empirical effectiveness of our approach. Regarding computing resources, all experiments included in this paper were conducted on NVIDIA A100 GPUs.

## 4.1. Gaussian Mixture Model

We evaluate URGE on a synthetic benchmark: a 30-dimensional Gaussian Mixture Model (GMM) with 40 equally weighted components. The target density is $p_{\text{data}}(x) = \frac{1}{40}\sum_{i=1}^{40}\mathcal{N}(x; \mu_i, 40I)$, where mean vectors $\mu_i$ are sampled uniformly from $[-40, 40]^{30}$. In this setting, both the score $\nabla \log p_t$ and the potential $\log p_t$ admit closed-form expressions, allowing performance to be evaluated without confounding score-estimation error.

The experiment adopts a reward-tilted setup. Particles are initialized as $X_0 \sim \mathcal{N}(0, I)$ and propagated using the guided dynamics in Eq. (2). We impose a quadratic reward, which is detailed together with other experimental settings in Appendix C.

Several guidance strategies are compared: Pure Guidance (PG) without control drift or resampling; AFDPS, which applies ESS-based resampling to PG dynamics; FK-Steering and URGE, both using $G = r$ with resampling; and a variance-controlling guidance (VCG) (Ren et al., 2025a) scheme that learns a control drift via regularized weighted least squares on top of AFDPS.

Performance is measured using Maximum Mean Discrepancy (MMD)(Smola et al., 2006), Sliced Wasserstein Dis-

| Method | MMD | SWD | Mean | Cov Frob |
|---|---|---|---|---|
| PG | 0.17 | 1.68 | 7.14 | 469.09 |
| AFDPS | 0.10 | 1.04 | 5.07 | 335.19 |
| AFDPS+VCG | 0.08 | 0.83 | 4.13 | 246.61 |
| FK-Steering | 0.07 | 0.85 | 4.86 | 198.20 |
| **URGE** | **0.06** | **0.62** | **3.20** | **181.31** |

*Table 2.* Performance of different strategies on the Gaussian mixture toy example. Lower values indicate closer alignment with the analytical reference.

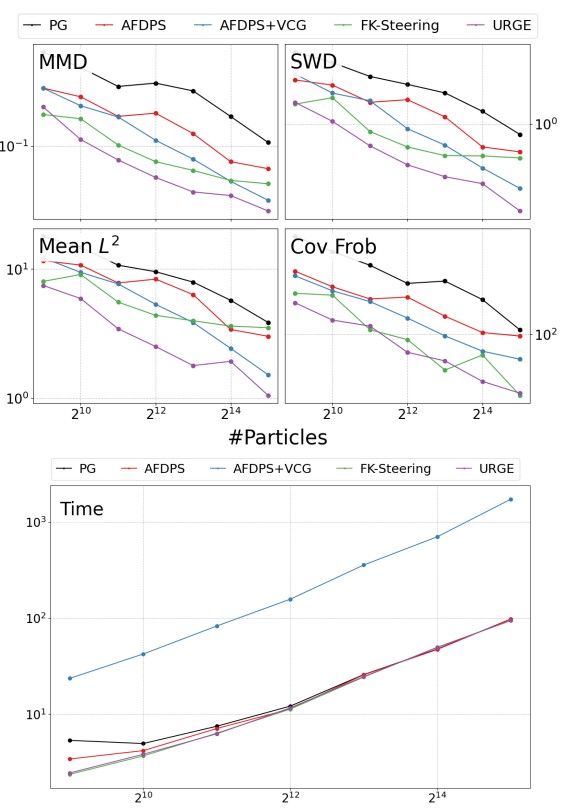

*Figure 2.* Performance metrics (top) and computational time (bottom) versus number of particles on Gaussian Mixture Model. Our URGE method surpasses all previous baselines on the analytical task.

tance (SWD) (Bonneel et al., 2015), mean vector $\ell_2$ error (Mean $L^2$), and covariance Frobenius error (Cov Frob). In table 2, URGE is tested on 5 random seeds and we report the averages. URGE consistently attains the lowest error across all metrics, demonstrating superior fidelity to the ground-truth distribution compared to all baselines. Notably, URGE surpasses AFDPS+VCG, achieving more robust convergence across various guidance regimes without the need for explicit variance control.

We also evaluate URGE across different particle counts while keeping the random seed fixed to isolate the effect of particle scale on both accuracy and runtime. Figure 2 illustrates the scaling behavior: as the number of particles increases, URGE not only outperforms the baselines but also converges more efficiently, achieving better results with fewer particles. Moreover, URGE exhibits competitive computational scaling compared to the baselines, especially AFDPS+VCG. These results confirm that URGE strikes an optimal balance between estimation fidelity and computational overhead, maintaining a practical runtime even as the number of particles scales. Moreover, we study the effect of dimensionality and discretization steps in Appendix D.

### 4.2. Inverse Problems

To ensure a rigorous comparison with AFDPS (Chen et al., 2025), we adopt the identical inverse-problem benchmarks and evaluation protocols established in this work. We use $\nabla \log p_t$ as the reward function in the experiment. Table 3 reports PSNR (Wang et al., 2004) and LPIPS (Zhang et al., 2018b) for four representative tasks on ImageNet-256. The tabulated results allow a head-to-head comparison of reconstruction fidelity and perceptual quality under identical evaluation protocols.

As shown in Table 3, URGE achieves performance comparable to AFDPS variants, while both significantly outperform other baselines. In addition, URGE achieves these results without requiring high-order derivatives, underscoring its efficiency and robustness in high-dimensional reconstruction tasks. Additional results of the inverse-problem experiment and visual examples are provided in the Appendix D.

### 4.3. Text-to-Image Generation

In text-to-image generation, reward models are typically complex neural networks whose gradients with respect to the sampling trajectory are either inaccessible or computationally prohibitive. Consequently, derivative-based methods like FK-Corrector and AFDPS are inapplicable in this setting. In our experiment, text-to-image generation is carried out using Stable Diffusion (Podell et al., 2023; Rombach et al., 2022), modeled as $p_{\text{data}}(x \mid c)$ for prompt $c$. The base model acts as the proposal generator. We consider publicly available diffusion models fine-tuned for prompt alignment and aesthetic preference. In particular, we consider SD v1.5 and SDXL, along with their DPO-tuned variants (Wallace et al., 2024; Rafailov et al., 2023), as the reward for prompt alignment.

Comparisons are conducted against the base model and FK-Steering. We focus on two aspects: aesthetic quality and qualitative prompt fidelity. The aesthetic alignment is measured using CLIP-Score (Hessel et al., 2021), ImageReward (Xu et al., 2023) and Human Preference Score (HPS) (Wu et al., 2023b). Table 4 reports averages over 50 prompts, each evaluated with three random seeds. To enhance the baseline, we use the best-of-n (bon) metrics

| Method | Gaussian Deblurring | | Motion Deblurring | | Super Resolution | | Box Inpainting | |
|---|---|---|---|---|---|---|---|---|
| | PSNR↑ | LPIPS↓ | PSNR↑ | LPIPS↓ | PSNR↑ | LPIPS↓ | PSNR↑ | LPIPS↓ |
| SGS-EDM | 22.09 | 0.4827 | 20.50 | 0.5255 | 15.43 | 0.6190 | 21.43 | 0.2977 |
| FK-Corrector | 18.36 | 0.5986 | 18.37 | 0.6006 | 18.58 | 0.5892 | 16.25 | 0.7137 |
| AFDPS-SDE | 22.43 | 0.3913 | 18.52 | 0.5196 | **21.03** | 0.4598 | 23.13 | 0.3065 |
| AFDPS-ODE | **22.57** | 0.4583 | **21.46** | **0.5030** | 19.60 | 0.5670 | 22.75 | **0.2748** |
| **URGE** | 22.38 | **0.3861** | 18.37 | 0.5252 | 21.00 | **0.4598** | **23.27** | 0.3054 |

*Table 3.* Results on 4 inverse problems for 100 validation images from ImageNet-256.

| Sampler | Clip-Score ↑ | HPS ↑ | IR ↑ | GenEval ↑ |
|---|---|---|---|---|
| $N = 1$ | 0.2730 | 0.2619 | 0.2144 | 0.6400 |
| $\nabla(N = 1)$ | 0.2726 | 0.2617 | 0.2074 | 0.6400 |
| FK ($N = 4$) | 0.2901 | 0.2849 | 0.8397 | 0.7200 |
| $\nabla(\text{FK}, N = 4)$ | 0.2899 | 0.2839 | 0.7906 | 0.7467 |
| **URGE** ($N = 4$) | **0.2997** | **0.2927** | **0.9955** | **0.7800** |

*Table 4.* Comparison of different sampling strategies for text-to-image generation. The $N = 1$ and $\nabla(N = 1)$ rows report best-of-n results on 10 random seeds. URGE achieves higher scores on all three metrics than the the base model, FK-Steering, and their gradient guidance variants, indicating the best overall quality.

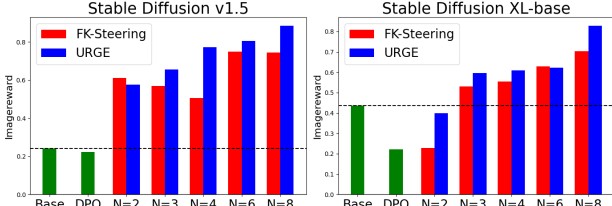

*Figure 3.* URGE achieves better performance compared with baseline, while exhibiting monotonic and larger reward increase with $N$ (unlike FK-Steering). With URGE, a smaller model (SD v1.5) can attain higher reward than an XL model.

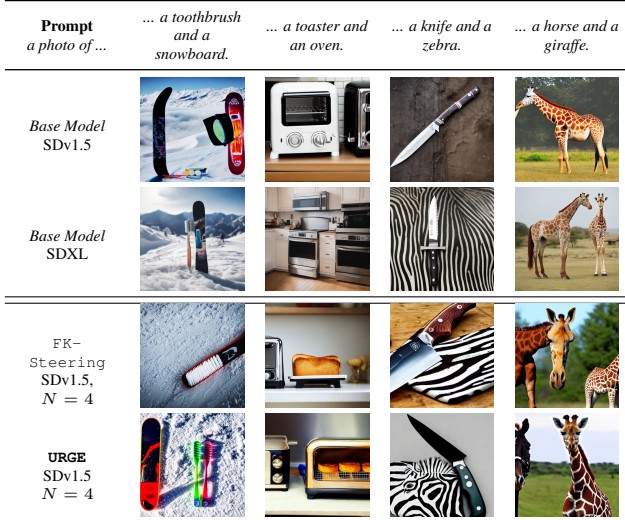

*Table 5.* Comparison of three methods under four prompts. URGE, even using SDv1.5, often matches or approaches the effect of the baseline that uses SDXL, and clearly outperforms FK-Steering, with results that better match the text and common sense.

for the rows $N = 1$ and $\nabla(N = 1)$, where the experiment is carried out on 10 random seeds and the maximum values of each metric were displayed on these seeds. With $N = 4$ particles, URGE consistently improves ImageReward scores over the baseline sampler, FK-Steering, and their gradient-guidance variants, indicating superior and more stable performance across diverse prompts.

Figure 3 examines particle-scaling effects on 10 prompts using ImageReward as the metric, with base and DPO-tuned models as baselines. As particle counts grow, URGE not only benefits from ensemble effects, but also consistently

surpasses FK-Steering. This indicates that URGE performs well on more particles and small models, which is valuable in practical applications with limited resources.

Table 5 provides qualitative comparisons under prompts drawn from GenEval (Ghosh et al., 2023). Because current models often struggle with prompts containing two objects (Cao et al., 2025), the selected prompts involve exactly two entities to stress-test compositional capabilities. Even when restricted to SD v1.5, URGE frequently matches or approaches outputs from the stronger SDXL baseline and consistently surpasses FK-Steering. Generated images show closer adherence to prompt semantics and common-sense consistency, demonstrating the practical benefits of URGE in real text-to-image settings. More experiment results are presented in the Appendix D.

## 5. Conclusion

We introduce URGE, a training-free, approximation-free ensemble-based posterior sampling framework that couples SMC with diffusion processes via a pathwise Girsanov reweighting mechanism. This formulation eliminates the need for score or Laplacian information of the reward and unifies path-based and state-based weighting by recovering AFDPS / FK-Corrector in the terminal-conditioned setting. Empirically, URGE consistently outperforms concurrent methods, achieving stronger aesthetic fidelity and improved prompt-faithful visualizations.

## Impact Statement

This paper presents work whose goal is to advance the field of machine learning. There are many potential societal consequences of our work, none of which we feel must be specifically highlighted here.

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

# A. Mathematical Formulation of Continuous-Time URGE

In this section, we reformulate the discrete-time URGE algorithm in an abstract framework and show how it converges, in the limit, to a continuous-time dynamical system. Specifically, we derive the continuous-time limit of URGE, which will play a key role in the proof of Theorem 3.1.

## A.1. Continuous-Time Jump-Diffusion System

As described in Section 2.4, for an arbitrary time interval $[s, t]$, we specify the path weight in (6) as follows:

$$w_{s,t}^{\text{URGE}}(X_{s:t}^G) = \exp\left(\int_s^t -V(\tau)\nabla_x G(X_\tau^G, \tau)^\top dW_\tau - \frac{1}{2}\int_s^t V^2(\tau)\|\nabla_x G(X_\tau^G, \tau)\|_2^2 d\tau + \int_s^t dr(X_\tau^G, \tau)\right),$$

which can be rewritten as

$$\log w_{s,t}^{\text{URGE}}(X_{s:t}^G) = \int_s^t -V(\tau)\nabla_x G(X_\tau^G, \tau)^\top dW_\tau - \frac{1}{2}\int_s^t V^2(\tau)\|\nabla_x G(X_\tau^G, \tau)\|_2^2 d\tau + \int_s^t dr(X_\tau^G, \tau). \tag{9}$$

In the implementation of URGE, we simulate $N$ independent trajectories. In the absence of resampling events, particles and weights evolve continuously: the particles follow the guided SDE, and the weights encode the Radon-Nikodym derivative between the base and guided path measures:

$$dX_t^{(i),G} = \left(v(X_t^{(i),G}, t) + V^2(t)\nabla_x G(X_t^{(i),G}, t)\right)dt + V(t)dW_t^{(i)}, \tag{10a}$$

$$d\log w_t^{(i)} = dr(X_t^{(i),G}, t) - V(t)\nabla_x G(X_t^{(i),G}, t)^\top dW_t^{(i)} - \frac{1}{2}V^2(t)\|\nabla_x G(X_t^{(i),G}, t)\|_2^2 dt, \tag{10b}$$

with initial weights $w_0^{(i)} = 1$ and $dr(X_t^{(i)}, t)$ denotes the Itô differential of the process $r(X_t^{(i)}, t)$ for $i \in [N]$. One should notice that (10b) derives immediately from (9).

Assumption A.1 on the guided diffusion

$$dX_t = \left(v(X_t) + V^2(t)\nabla_x G(X_t)\right)dt + V(t)dW_t$$

discussed in Section 3 ensures well-posedness and sufficient smoothness of the resulting processes. The constraints on the reward serves to make the coefficients of (10) Lipschitz and the usage of Girsanov's Theorem legal (Novikov's condition). This condition matches the classical regularity assumptions on diffusion coefficients described in (Ethier and Kurtz, 1986; Ikeda and Watanabe, 1981).

> **Assumption A.1.** Fix $T > 0$. The drift $v : [0, T] \times \mathbb{R}^d \to \mathbb{R}^d$, the guidance $G : [0, T] \times \mathbb{R}^d \to \mathbb{R}$, the reward $r : [0, T] \times \mathbb{R}^d \to \mathbb{R}$ the diffusion scalar $V : [0, T] \to \mathbb{R}$ satisfy:
>
> 1. $r, G \in C^{1,2}(\mathbb{R}^d \times [0, T])$, and $\partial_t r, \nabla_x r, \nabla_x^2 r, \partial_t G, \nabla_x G, \nabla_x^2 G$ are bounded on $\mathbb{R}^d \times [0, T]$.
>
> 2. $v(\cdot, t)$ and $\nabla_x G(\cdot, t)$ are globally Lipschitz in $x$, uniformly in $t \in [0, T]$, i.e., there exist constants $L, K > 0$ such that
>    $$\|v(x, t) - v(y, t)\| + \|\nabla_x G(x, t) - \nabla_x G(y, t)\| \leq L\|x - y\|$$
>    and
>    $$\|v(x, t)\| + \|\nabla_x G(x, t)\| \leq K(1 + \|x\|),$$
>    for any $x, y \in \mathbb{R}^d, t \in [0, T]$.
>
> 3. $V$ is measurable and uniformly lower- and upper- bounded, i.e.,
>    $$0 < V_{\min} \leq V(t) \leq V_{\max}, \quad \forall t \in [0, T].$$
>
> 4. There exists a constant $L_r > 0$ such that
>    $$\left\|\nabla_x r(y, t)^\top v(y, t) - \nabla_x r(x, t)^\top v(x, t)\right\| \leq L_r\|y - x\|, \quad \forall x, y \in \mathbb{R}^d, \forall t \in [0, T].$$

## A.2. Continuous-Time Limit of URGE

Lemma A.2 formalizes the resampling step by characterizing ancestor selection through inverse transform sampling. Specifically, given normalized weights

$$\widehat{\boldsymbol{w}} = (\widehat{w}^{(1)}, \ldots, \widehat{w}^{(N)}), \quad \widehat{w}^{(i)} = \frac{w^{(i)}}{\sum_{j=1}^{N} w^{(j)}},$$

and a mark $v \sim \mathrm{Unif}[0,1]$, we define the ancestor selection map $\mathcal{A} : \mathbb{R}^N \times [0,1] \to \{1, \ldots, N\}$ as

$$\mathcal{A}(\widehat{\boldsymbol{w}}, v) := \min\left\{ k : \sum_{i=1}^{k} \widehat{w}^{(i)} \geq v \right\}.$$

Applying this map component-wise to $\boldsymbol{v}$ yields parent indices for all particles, with selection probabilities proportional to the normalized weights.

**Lemma A.2.** *For normalized weights $\widehat{\boldsymbol{w}} = (\widehat{w}^{(1)}, \ldots, \widehat{w}^{(N)})$ with $\widehat{w}^{(i)} = w^{(i)} / \sum_{j=1}^{N} w^{(j)}$ and $v \sim \mathrm{Unif}[0,1]$, then the mapping $\mathcal{A}$ defined as above coincides with the multinomial resampling rule used in the jump component of URGE, i.e.*

$$\mathbb{P}\left(\mathcal{A}(\widehat{\boldsymbol{w}}, v) = k\right) = \widehat{w}^{(k)}.$$

*Proof of Lemma A.2.* Define $S_0 = 0$, and $S_k = \sum_{i=1}^{k} \widehat{w}^{(i)}$. By definition of $\mathcal{A}$, $\mathcal{A}(\widehat{\boldsymbol{w}}, v) = k$ equals that $w^{(k)}$ is the first index such that $S_k \geq v$. Equivalently, $S_{k-1} < v \leq S_k$. By $v \sim U[0,1]$, we have

$$\mathbb{P}(S_{k-1} < v \leq S_k) = \widehat{w}^{(k)},$$

which turns out that $\mathbb{P}(\mathcal{A}(\widehat{\boldsymbol{w}}, v) = k) = \widehat{w}^{(k)}$. □

We now specify the Lévy-driven jump component of the dynamics. Let $N_t^{\mathrm{res}}$ denote the counting process recording the number of resampling events in $[0, t]$. Its intensity is given by $\Lambda(\mu_{t-}^N)$, where

$$\mu_t^N := \sum_{i=1}^{N} \widehat{w}^{(i)} \delta_{X_t^{(i),G}}$$

denotes the empirical measure. $\Lambda : \mathcal{P}(\mathbb{R}^d) \to \mathbb{R}_+$ is a measurable functional defined on the space of Borel probability measures. For the detailed definition and discussions on the Poisson random measure, we refer readers to previous literatures (Applebaum, 2009; Ren et al., 2024; 2025b).

We make the following assumption on the resampling procedure.

**Assumption A.3.** $\Lambda$ is uniformly bounded, *i.e.*, there exists a constant $K < \infty$ such that $\Lambda(\mu) \leq K$ for all $\mu \in \mathcal{P}(\mathbb{R}^d)$.

Assumption A.3 is a standard regularity condition in particle methods with state-dependent jump intensities. Under this assumption, the associated resampling process is well-posed and non-explosive, in the sense that only finitely many jumps can occur over any finite time horizon. Similar bounded intensity conditions have been employed in the literature on importance sampling for Markovian intensity models (Glasserman and Merener, 2004; Djehiche et al., 2022; Miasojedow and Niemiro, 2016), where the boundedness of the intensity plays a key role in establishing the existence and stability of the sampling scheme. One practical interpretation of this assumption in the context of SMC is that it precludes pathological regimes in which the Effective Sample Size (ESS) collapses excessively, thereby facilitating stable resampling behavior.

Randomized resampling can be modeled via a Poisson random measure $\mathcal{N}(\mathrm{d}t, \mathrm{d}u, \mathrm{d}\boldsymbol{v})$ on $[0, \infty) \times [0, \infty) \times [0, 1]^N$. Here $t$ denotes time, $u$ the jump-intensity signal, and $\boldsymbol{v} \in [0, 1]^N$ the resampling marks that determine which particle is copied at

a jump. Each coordinate of $\boldsymbol{v}$ is drawn from the uniform distribution $\mathrm{Unif}[0, 1]$, and the mean intensity measure is Lebesgue, *i.e.*,

$$\mathbb{E}[\mathcal{N}(\mathrm{d}t, \mathrm{d}u, \mathrm{d}\boldsymbol{v})] = \mathrm{d}t\mathrm{d}u\mathrm{d}\boldsymbol{v}.$$

Using the Poisson random measure $\mathcal{N}(\mathrm{d}t, \mathrm{d}u, \mathrm{d}\boldsymbol{v})$, the process admits the representation

$$N_t^{\mathrm{res}} = \int_0^t \int_0^\infty \int_{[0,1]^N} \mathbf{1}_{\{u \leq \Lambda(\mu_{s-}^N)\}} \mathcal{N}(\mathrm{d}s, \mathrm{d}u, \mathrm{d}\boldsymbol{v}).$$

Define the weight mean $\overline{w}_t = \frac{1}{N} \sum_{i=1}^N w_t^{(i)}$. The resulting continuous-time jump-diffusion system is

$$\mathrm{d}X_t^{(i),G} = \left(v(X_t^{(i),G}, t) + V^2(t)\nabla_x G(X_t^{(i),G}, t)\right)\mathrm{d}t + V(t)\mathrm{d}W_t^{(i)}$$
$$+ \int_0^\infty \int_{[0,1]^N} \left(X_{t-}^{(\mathcal{A}(\widehat{\boldsymbol{w}}_{t-}, v_i)),G} - X_{t-}^{(i),G}\right)\mathbf{1}_{\{u \leq \Lambda(\mu_{t-}^N)\}} \mathcal{N}(\mathrm{d}t, \mathrm{d}u, \mathrm{d}\boldsymbol{v}),$$

$$\mathrm{d}\log w_t^{(i)} = \mathrm{d}r(X_t^{(i),G}, t) - V(t)\nabla_x G(X_t^{(i),G}, t)^\top \mathrm{d}W_t^{(i)} - \frac{1}{2}\|V(t)\nabla_x G(X_t^{(i),G}, t)\|_2^2 \mathrm{d}t$$
$$+ \int_0^\infty \int_{[0,1]^N} \left(\log \overline{w}_{t-} - \log w_{t-}^{(i)}\right)\mathbf{1}_{\{u \leq \Lambda(\mu_{t-}^N)\}} \mathcal{N}(\mathrm{d}t, \mathrm{d}u, \mathrm{d}\boldsymbol{v}).$$

Finally, we show that the proposed jump-diffusion dynamics arises as the weak limit of the discrete resampling algorithm. Direct analysis of the discrete-time scheme is facilitated by introducing a vanishingly sparse resampling mechanism that matches the intensity of the limiting jump process. Specifically, let

$$0 = t_0 < t_1 < \cdots < t_K = T$$

be a time grid with step size $\Delta t_k := t_k - t_{k-1}$ and

$$\Delta t := \max_{k=1,\ldots,K} \Delta t_k$$

be the maximum time step. At each grid point $t_k$ ($k = 1, \ldots, K-1$), resampling is triggered independently with probability $p_k = \Lambda(\mu_{t_k-}^N)\Delta t_k$. The resulting weak convergence to the continuous-time jump-diffusion process is established in Theorem A.4.

Define $X_t^{(i),G,\Delta t} := X_{t_k}^{i,G}, w_t^{(i),\Delta t} := w_{t_k}^{(i)}$ for $t \in [t_k, t_{k+1})$, where $t_k = k\Delta t$, and

> **Lemma A.4** (Continuous Time Limit of URGE). *Consider URGE particle system with time step $\Delta t > 0$, and resampling events occurring independently at each $t_k$ with probability $p_k = \Lambda(\mu_{t_k-}^N)\Delta t$. Then, as $\Delta t \to 0$, the $N$-particle process $\{X_t^{(i),G,\Delta t}, w_t^{(i),\Delta t}\}_{i=1}^N$ viewed as a random element of Skorokhod space $D([0,T]; (\mathbb{R}^d \times \mathbb{R})^{\times N})$ converges weakly to the continuous-time jump-diffusion particle system $\{X_t^{(i),G}, w_t^{(i)}\}_{i=1}^N$.*

To simplify the following discussion, we introduce the following notation: let $E = (\mathbb{R}^d \times \mathbb{R})^N$ be the state space of the particle system and let

$$\boldsymbol{z}_t = \left(X_t^{(1),G}, w_t^{(1)}, X_t^{(2),G}, w_t^{(2)}, \cdots, X_t^{(N),G}, w_t^{(N)}\right) \in E$$

denote the full particle state at time $t$, and let $C_c^2(E)$ be the space of twice continuously differentiable test functions with compact support.

For $f \in C_c^2(E)$, the discrete generator is

$$L_t^{\Delta t} f(\boldsymbol{z}) = \frac{1}{\Delta t}\left(\mathbb{E}[f(\boldsymbol{z}_{t+\Delta t}) \mid \boldsymbol{z}_t = \boldsymbol{z}] - f(\boldsymbol{z})\right).$$

Now we denote

$$\Psi(\boldsymbol{z}_t, \boldsymbol{v}) := (X_t^{(\mathcal{A}(\widehat{\boldsymbol{w}}_t, v_1)),G}, \log \overline{w}_t, X_t^{(\mathcal{A}(\widehat{\boldsymbol{w}}_t, v_2)),G}, \log \overline{w}_t, \cdots, X_t^{(\mathcal{A}(\widehat{\boldsymbol{w}}_t, v_N)),G}, \log \overline{w}_t)$$

as the resampling map of state and averaging procedure of weights discussed above and let $\mathcal{L}_t^{\mathrm{diff}}$ be the diffusion generator.

## A.3. Mathematical Properties of the Particle System

Before we prove the lemma, we first establish the necessary conditions for the weak convergence of the discrete-time particle system to its continuous-time limit. Specifically, we address generator convergence, well-posedness of the limit, and tightness.

> **Lemma A.5.** *The discrete-time generator $L_t^{\Delta t}$ converges uniformly to the generator $\mathcal{L}_t$ of the limiting continuous-time jump-diffusion on the core $C_c^2(E)$, defined as*
>
> $$\mathcal{L}_t f(\boldsymbol{z}) = \mathcal{L}_t^{\text{diff}} f(\boldsymbol{z}) + \Lambda(\mu_{t-}^N) \int_{[0,1]^N} (f(\Psi(\boldsymbol{z}, \boldsymbol{v})) - f(\boldsymbol{z})) \, d\boldsymbol{v}.$$

*Proof of Lemma A.5.* We prove the lemma in two steps.

**Step 1: Generator of the Continuous Process.** By Itô's formula,

$$dr(X_t^G, t) = \partial_t r(X_t^G, t)dt + \nabla_x r(X_t^G, t)^\top \left(v(X_t^G, t) + V^2(t)\nabla_x G(X_t^G, t)\right)dt$$
$$+ \frac{1}{2}V^2(t)\Delta_x r(X_t^G, t)dt + V(t)\nabla_x r(X_t^G, t)^\top dW_t.$$

We use a generalized version of dynamics of the continuous process $\boldsymbol{z}_t$:

$$d\boldsymbol{z}_t = v_{\text{diff}}(\boldsymbol{z}_t)dt + \sigma_{\text{diff}}(\boldsymbol{z}_t)dW_t + \int_0^\infty \int_{[0,1]^N} (\Psi(\boldsymbol{z}_{t-}, \boldsymbol{v}) - \boldsymbol{z}_{t-})\mathbf{1}_{\{u \leq \Lambda(\mu_{t-}^N)\}}\mathcal{N}(dt, du, d\boldsymbol{v}),$$

where

$$v_{\text{diff}}(\boldsymbol{z}_t, t) = \Bigg(v(X_t^{(i),G}, t) + V^2(t)\nabla_x G(X_t^{(i),G}, t), \partial_t r(X_t^G, t) + \nabla_x r(X_t^{(i),G}, t)^\top \left(v(X_t^{(i),G}, t) + V^2(t)\nabla_x G(X_t^{(i),G}, t)\right)$$
$$+ \frac{1}{2}V^2(t)\Delta_x r(X_t^{(i),G}, t) - \frac{1}{2}\|\theta(X_t^{(i),G}, t)\|^2\Bigg)_{i=1}^N,$$

and

$$\sigma_{\text{diff}}(\boldsymbol{z}_t, t) = \left(V(t), V(t)\nabla_x r(X_t^{(i),G}, t)^\top - V(t)\nabla_x G(X_t^{(i),G}, t)^\top\right)_{i=1}^N.$$

By applying Itô's formula for semimartingales with jumps, the time evolution of $f(\boldsymbol{z}_t)$ is given by:

$$f(\boldsymbol{z}_t) - f(\boldsymbol{z}_0) = \int_0^t \mathcal{L}_t^{\text{diff}} f(\boldsymbol{z}_s)ds + \int_0^t \nabla f^\top \sigma_{\text{diff}}dW_s$$
$$+ \int_0^t \int_0^\infty \int_{[0,1]^N} (f(\Psi(\boldsymbol{z}_{s-}, \boldsymbol{v})) - f(\boldsymbol{z}_{s-})) \mathbf{1}_{\{u \leq \Lambda(\mu_{s-}^N)\}}\mathcal{N}(ds, du, d\boldsymbol{v}),$$

where the generator for the diffusion part is defined as

$$\mathcal{L}_t^{\text{diff}} f(\boldsymbol{z}) = v_{\text{diff}}(\boldsymbol{z}, t)^\top \nabla f(\boldsymbol{z}) + \frac{1}{2}\text{tr}(\sigma_{\text{diff}}^2(\boldsymbol{z}, t)\nabla^2 f(\boldsymbol{z})).$$

Applying Itô's formula for semimartingales to a test function $f \in C_c^2(E)$, and introducing the compensated Poisson measure

$$\widetilde{\mathcal{N}}(ds, du, d\boldsymbol{v}) = \mathcal{N}(ds, du, d\boldsymbol{v}) - ds du d\boldsymbol{v},$$

we decompose the evolution of $f(\boldsymbol{z}_t)$ into drift and martingale components. Since integrals with respect to $W_t$ and $\widetilde{\mathcal{N}}$ are local martingales with vanishing expectations, the generator $\mathcal{L}$ is defined by the predictable drift terms:

$$\mathcal{L}_t f(\boldsymbol{z}) = \mathcal{L}_t^{\text{diff}} f(\boldsymbol{z}) + \int_0^\infty \int_{[0,1]^N} (f(\Psi(\boldsymbol{z}, \boldsymbol{v})) - f(\boldsymbol{z})) \mathbf{1}_{\{u \leq \Lambda(\mu_{t-}^N)\}}du d\boldsymbol{v}.$$

Evaluating the integral over the auxiliary variable $u$, we invoke the identity $\int_0^\infty \mathbf{1}_{\{u \le \Lambda(z)\}} \mathrm{d}u = \Lambda(z)$, yielding the continuous generator:

$$\mathcal{L}_t f(z) = \mathcal{L}_t^{\text{diff}} f(z) + \Lambda(\mu_{t-}^N) \int_{[0,1]^N} (f(\Psi(z, v)) - f(z)) \, \mathrm{d}v.$$

**Step 2: Generator of the Discrete Process.** We now analyze the discrete generator

$$L_t^{\Delta t} f(z) = \Delta t^{-1} \mathbb{E}[f(z_{t+\Delta t}) - f(z) \mid z_t = z].$$

The process evolves via diffusion with probability $1 - \Lambda(\mu_{t-}^N)\Delta t$ and undergoes resampling with probability $\Lambda(\mu_{t-}^N)\Delta t$.

For the diffusion step, Taylor expansion implies

$$\mathbb{E}_{\text{diff}}[f(z_{t+\Delta t})] = f(z) + \mathcal{L}_t^{\text{diff}} f(z)\Delta t + o(\Delta t).$$

For the resampling step, the expectation over the multinomial selection of indices is given by:

$$\mathbb{E}_{\text{res}}[f(z_{\text{new}})] = \sum_{k_1=1}^N \cdots \sum_{k_N=1}^N \left( \prod_{i=1}^N \widehat{w}^{(k_i)} \right) f\left(z(k_1, \ldots, k_N)\right).$$

where $z(k_1, \ldots, k_N)$ represents the state where the $i$-th particle is replaced by the $k_i$-th parent.

We now show this discrete sum is equivalent to the integral representation used in the continuous generator. Consider the auxiliary variables $v = (v_1, \ldots, v_N) \in [0, 1]^N$. The mapping $\Psi(z, v)$ assigns particle $i$ to parent $k$ if $v_i$ falls into the $k$-th cumulative weight interval $I_k = (\sum_{i=1}^{k-1} \widehat{w}^{(i)}, \sum_{i=1}^k \widehat{w}^{(i)}]$, where $|I_k| = \widehat{w}^{(k)}$. Since $v_1, \ldots, v_N$ are independent uniform random variables, the integral over the hypercube factors as:

$$\int_{[0,1]^N} f(\Psi(z, v)) \mathrm{d}v = \int_0^1 \cdots \int_0^1 f\left(z\left(\sum_{k_1} k_1 \mathbf{1}_{I_{k_1}}(v_1), \ldots, \sum_{k_N} k_N \mathbf{1}_{I_{k_N}}(v_N)\right)\right) dv_1 \ldots dv_N$$

$$= \sum_{k_1=1}^N \cdots \sum_{k_N=1}^N \left( \int_{I_{k_1}} dv_1 \cdots \int_{I_{k_N}} dv_N \right) f\left(z(k_1, \ldots, k_N)\right)$$

$$= \sum_{k_1=1}^N \cdots \sum_{k_N=1}^N \left( \prod_{i=1}^N \widehat{w}^{(k_i)} \right) f\left(z(k_1, \ldots, k_N)\right).$$

This establishes the following identity:

$$\mathbb{E}_{\text{res}}[f(z_{\text{new}})] = \int_{[0,1]^N} f(\Psi(z, v)) \mathrm{d}v.$$

Substituting the results back into the expression for $L_t^{\Delta t} f(z)$:

$$L_t^{\Delta t} f(z) = \frac{1}{\Delta t} \left[ (1 - \Lambda(\mu_{t-}^N)\Delta t)(f(z) + \mathcal{L}_{\text{diff}} f(z)\Delta t) + \Lambda(\mu_{t-}^N)\Delta t \int_{[0,1]^N} f(\Psi(z, v))\mathrm{d}v - f(z) + o(\Delta t) \right]$$

$$= \mathcal{L}_t^{\text{diff}} f(z) - \Lambda(\mu_{t-}^N) f(z) + \Lambda(\mu_{t-}^N) \int_{[0,1]^N} f(\Psi(z, v))\mathrm{d}v + o(1)$$

$$= \mathcal{L}_t^{\text{diff}} f(z) + \Lambda(\mu_{t-}^N) \int_{[0,1]^N} (f(\Psi(z, v)) - f(z)) \, \mathrm{d}v + o(1).$$

Taking the limit as $\Delta t \to 0$, we obtain

$$\lim_{\Delta t \to 0} L_t^{\Delta t} f(z) = \mathcal{L}_t f(z).$$

Since the generator of the discrete approximation converges uniformly to the generator of the continuous jump-diffusion process, the discrete system converges weakly to the continuous system. $\qquad\square$

**Lemma A.6.** *The stochastic system* (10a) *has a unique strong solution $z_t$ on $t \in [0, T]$.*

*Proof of Lemma A.6.* Denote $\tau_0 = 0$, and $\tau_i > 0$ to be the time of $i$-th jump. Under our regularity assumptions on $v, V, r, G$, the drift term $v_{\text{diff}}$ and the diffusion coefficient $\sigma_{\text{diff}}$ are Lipschitz in the whole path space. Due to classic existence-uniqueness theorem of SDEs, the system

$$\mathrm{d}z_t = v_{\text{diff}}(z_t)\mathrm{d}t + \sigma_{\text{diff}}(z_t)\mathrm{d}W_t$$

has a unique strong solution $y_t^{(0)}$ on $[\tau_0, +\infty)$.

The first jump moment $\tau_1$ can be expressed as

$$\tau_1 = \inf \left\{ t > 0 : \int_0^t \int_0^\infty \int_{[0,1]^N} \mathbf{1}_{\{u \leq \Lambda(Y_s^{(0)})\}} \mathcal{N}(\mathrm{d}s, \mathrm{d}u, \mathrm{d}v) \geq 1 \right\}.$$

Note that since $\Lambda$ is uniformly bounded, $\tau_1 > 0$ a.s. We define the solution in $t \in [0, \tau_1)$ as $z_t = y_t^{(0)}$.

At time $\tau_1$, we process a jump. The state right before jump is $z_{\tau_1-} = y_{\tau_1-}^{(0)}$, and the after-jump state is defined as $z_{\tau_1} = \Psi(z_{\tau_1-}, v_{\tau_1})$, where $v_{\tau_1}$ is the mark variable the Poisson measure sampled at $\tau_1$. Since $z_{\tau_1-}$ is $\mathcal{F}_{\tau_1-}$ measurable, and $v_{\tau_1-}$ is a determined variable at time $\tau_1$, thus $z_{\tau_1}$ is well-defined.

We do induction on the jump count $k \geq 1$. Suppose we have already construct the unique strong solution $z_t$ up to the $k$-th jump moment. Using $z_{\tau_k}$ as the new initials, the diffusion process has a unique strong solution $y_t^{(k)}$. The next jump moment is defined as

$$\tau_{k+1} = \inf \left\{ t > \tau_k : \int_{\tau_k}^t \int_0^\infty \int_{[0,1]^N} \mathbf{1}_{\{u \leq \Lambda(Y_s^{(k)})\}} \mathcal{N}(\mathrm{d}s, \mathrm{d}u, \mathrm{d}v) \geq 1 \right\}.$$

On the interval $[\tau_k, \tau_{k+1})$, we define the solution as $z_t = y_t^{(k)}$; and at time $\tau_{k+1}$, define $z_{\tau_{k+1}} = \Psi(z_{\tau_{k+1}-}, v_{\tau_{k+1}})$.

Therefore, the strong solution can be defined on $[0, \tau_\infty)$, where $\tau_\infty := \lim_{k\to\infty} \tau_k$. In order to show that a unique strong solution exists on $[0, T]$, we need to show that $\tau_\infty = \infty$ a.s. Suppose $\Lambda$ is uniformly bounded by some constant $K$. Denote $N_t^K$ as a homogeneous Poisson process with intensity $K$. By stochastic domination, we have

$$\mathbb{P}(\tau_k \leq T) \leq \mathbb{P}(N_T^K \geq k).$$

Since $N_T^K < \infty$ a.s., we derive that $\lim_{k\to\infty} P(\tau_k \leq T) = 0$, which means that $\tau_k \to \infty$ as $k \to \infty$.

Combining the above discussion, we have that for all finite $T$, we can uniquely construct the strong solution of (10a) by finite-step induction. $\square$

**Lemma A.7.** $\{X^{(i),G,\Delta t}, w^{(i),\Delta t}\}_{i=1}^N$ *is tight in the Skorokhod space* $\mathcal{D}([0,T]; (\mathbb{R}^d \times \mathbb{R})^N)$.

*Proof of Lemma A.7.* Let

$$z_t^{\Delta t} = (X_t^{(1),G,\Delta t}, w_t^{(1),\Delta t}, X_t^{(2),G,\Delta t}, w_t^{(2),\Delta t}, \cdots, X_t^{(N),G,\Delta t}, w_t^{(N),\Delta t}) \in E = (\mathbb{R}^d \times \mathbb{R})^N$$

denotes the state after the diffusion increment but before the resampling jump.

We use Aldous-Billingsley tightness criterion (Corollary under Theorem 13.2 of (Billingsley, 1999)) for the proof of tightness. Specifically, we focus on the following two properties:

**Step 1: Pointwise Moment Bound.** We need to prove $\limsup_{\Delta t > 0} \sup_k \mathbb{E}[\|X_{t_k}\|^2] < \infty$. In the Euler-Maruyama step,

$$z_{t_{k+1}-}^{\Delta t} = z_{t_k}^{\Delta t} + v_{\text{diff}}(z_{t_k}^{\Delta t}, t_k)\Delta t + \sigma_{\text{diff}}(z_{t_k}^{\Delta t}, t_k)\sqrt{\Delta t}\xi_{t_k},$$

where $\xi_{t_k} \sim \mathcal{N}(0, I_{(d+1)N})$. Under regularity assumptions, there exists constants $C > 0$ independent of time $t_k$ such that

$$\mathbb{E}[\|z_{t_{k+1}-}^{\Delta t}\|^2 \mid z_{t_k}^{\Delta t}] = \|z_{t_k}^{\Delta t}\|^2 + 2(z_{t_k}^{\Delta t})^\top v_{\text{diff}}(z_{t_k}^{\Delta t}, t)\Delta t + 2\mathbb{E}[(z_{t_k}^{\Delta t})^\top \sigma_{\text{diff}}(z_{t_k}^{\Delta t}, t_k)\sqrt{\Delta t}\xi_{t_k} \mid z_{t_k}^{\Delta t}]$$

$$+ \mathbb{E}[(\sigma_{\mathrm{diff}}(z_{t_k}^{\Delta t}, t_k)\sqrt{\Delta t}\xi_{t_k})^2 \mid z_{t_k}^{\Delta t}] + o(\Delta t)$$

$$\leq \|z_{t_k}^{\Delta t}\|^2 + (\|z_{t_k}^{\Delta t}\|^2 + \|v_{\mathrm{diff}}(z_{t_k}^{\Delta t}, t_k)\|^2 + \|\sigma_{\mathrm{diff}}(z_{t_k}^{\Delta t}, t_k)\|^2)\Delta t + o(\Delta t)$$

$$\leq \|z_{t_k}^{\Delta t}\|^2(1 + C\Delta t) + C\Delta t.$$

In the resampling step, the conditional second moment is given by

$$\mathbb{E}[\|z_{t_{k+1}}^{\Delta t}\|^2 \mid z_{t_{k+1}-}^{\Delta t}] = \sum_{i=1}^{N} \widehat{w}_{t_k}^{(i)} \|X_{t_{k+1}-}^{(i),G,\Delta t}\|^2 + N \log \overline{w}_{t_{k+1}-}$$

$$\leq N \left( \max_{1 \leq i \leq N} \|X_{t_{k+1}-}^{(i),G,\Delta t}\|^2 + \max_{1 \leq i \leq N} \log w_{t_{k+1}-}^{(i)} \right)$$

$$\leq N \|z_{t_{k+1}-}^{\Delta t}\|^2.$$

Counting in the jump intensity $p_k = \Lambda(\mu_{t_{k+1}-}^N)\Delta t$, we have that

$$\mathbb{E}[\|z_{t_{k+1}}^{\Delta t}\|^2] \leq KN\Delta t \mathbb{E}[\|z_{t_{k+1}-}^{\Delta t}\|^2],$$

Finally, by Discrete Gronwall's Lemma, we get

$$\sup_{0 \leq k \leq T/\Delta t} \mathbb{E}[\|z_{t_k}^{\Delta t}\|^2] \leq \left( \mathbb{E}[\|z_0^{\Delta t}\|^2] + C'T \right)(1 + C'\Delta t)^{\frac{T}{\Delta t}} \leq \left( \mathbb{E}[\|z_0^{\Delta t}\|^2] + C'T \right)e^{C'T}.$$

for some constant $C'$ depending only on bound on $r, G, K, N$, which coincides with our claim.

**Step 2: Aldous Criterion.** We verify the Aldous criterion for tightness in $D([0,T]; \mathbb{R}^d)$. Let $\tau$ be any stopping time with respect to the filtration generated by $z_t^{\Delta t}$, bounded by $T$. Let $\delta > 0$ and $\varepsilon > 0$. We aim to show that:

$$\lim_{\delta \downarrow 0} \limsup_{\Delta t \to 0} \mathbb{P}\left( \|z_{\tau+\delta}^{\Delta t} - z_\tau^{\Delta t}\| > \varepsilon \right) = 0.$$

Since the process $z_t^{\Delta t}$ is piecewise constant, the value at time $s$ corresponds to the discrete state $z_{t_k}^{\Delta t}$ with $t_k \leq s < t_{k+1}$. The increment over the interval $[\tau, \tau + \delta]$ involves summing the discrete updates over the time indices

$$\mathcal{K}_{\tau,\delta} := \{k : \tau \leq t_k < \tau + \delta\}.$$

The number of steps in this interval is bounded by $\lceil \delta/\Delta t \rceil$.

We decompose the total increment into three components: the cumulative continuous drift $\mathcal{A}_{\tau,\delta}$, the cumulative diffusive noise $\mathcal{M}_{\tau,\delta}$, and the cumulative resampling jumps $\mathcal{J}_{\tau,\delta}$:

$$z_{\tau+\delta}^{\Delta t} - z_\tau^{\Delta t} = \underbrace{\sum_{k \in \mathcal{K}_{\tau,\delta}} v_{\mathrm{diff}}(z_{t_k}^{\Delta t}, t_k)\Delta t}_{\mathcal{A}_{\tau,\delta}} + \underbrace{\sum_{k \in \mathcal{K}_{\tau,\delta}} \sigma_{\mathrm{diff}}(z_{t_k}^{\Delta t}, t_k)\sqrt{\Delta t}\xi_{t_k}}_{\mathcal{M}_{\tau,\delta}} + \underbrace{\sum_{k \in \mathcal{K}_{\tau,\delta}} (z_{t_{k+1}}^{\Delta t} - z_{t_{k+1}-}^{\Delta t})}_{\mathcal{J}_{\tau,\delta}}.$$

By the triangle inequality, it suffices to bound the probability of each term exceeding $\varepsilon/3$. We rely on the pointwise moment bound uniformly on $t_i, i = 1, 2, \cdots, T/\Delta t$ in the previous passage.

1. **Drift Term $\mathcal{A}_{\tau,\delta}$.** Using Markov's inequality and the linear growth condition $\|v_{\mathrm{diff}}(z, t)\| \leq K(1 + \|z\|)$ for all $z \in \mathbb{R}^{(d+1) \times N}, t \in [0, T]$, we obtain

$$\mathbb{P}(\|\mathcal{A}_{\tau,\delta}\| > \frac{\varepsilon}{3}) \leq \frac{3}{\varepsilon} \mathbb{E}\left[ \sum_{k \in \mathcal{K}_{\tau,\delta}} \|v_{\mathrm{diff}}(z_{t_k}^{\Delta t}, t_k)\|\Delta t \right].$$

Since the interval length is at most $\delta + \Delta t$, we have:

$$\mathbb{E}\left[ \sum_{k \in \mathcal{K}_{\tau,\delta}} \|v_{\mathrm{diff}}(z_{t_k}^{\Delta t}, t_k)\|\Delta t \right] \leq (\delta + \Delta t) \sup_k \mathbb{E}[K_1(1 + \|z_{t_k}^{\Delta t}\|)] \leq C_1(\delta + \Delta t).$$

$K, C_1$ are constants independent of time. Thus, we have that

$$\limsup_{\Delta t \to 0} \mathbb{P}(\|\mathcal{A}_{\tau,\delta}\| > \varepsilon/3) \leq \frac{3C_1\delta}{\varepsilon},$$

which vanishes as $\delta \downarrow 0$.

2. **Diffusion Term $\mathcal{M}_{\tau,\delta}$.** The term $\mathcal{M}_{\tau,\delta}$ approximates a continuous martingale. Using Chebyshev's inequality and the orthogonality of martingale increments, we have

$$\mathbb{P}(\|\mathcal{M}_{\tau,\delta}\| > \frac{\varepsilon}{3}) \leq \frac{9}{\varepsilon^2} \mathbb{E}\left[\left\|\sum_{k \in \mathcal{K}_{\tau,\delta}} \sigma_{\text{diff}}(z_{t_k}^{\Delta t}, t_k)\sqrt{\Delta t}\xi_k\right\|^2\right]$$

$$= \frac{9}{\varepsilon^2} \sum_{k \in \mathcal{K}_{\tau,\delta}} \mathbb{E}[\|\sigma_{\text{diff}}(z_{t_k}^{\Delta t}, t_k)\|^2]\Delta t$$

$$\leq \frac{9}{\varepsilon^2} \sup_{k \in \mathcal{K}_{\tau,\delta}} \mathbb{E}[K_2(1 + \|z_{t_k}^{\Delta t}\|)](\delta + \Delta t) \leq \frac{9C_2(\delta + \Delta t)}{\varepsilon^2}$$

where the second last inequality is by the pointwise moment bound and the boundedness of $V$. Thus, we have that

$$\limsup_{\Delta t \to 0} \mathbb{P}(\|\mathcal{M}_{\tau,\delta}\| > \varepsilon/3) \leq \frac{9C_2\delta}{\varepsilon^2},$$

which vanishes as $\delta \downarrow 0$.

3. **Resampling Jump Term $\mathcal{J}_{\tau,\delta}$.** The term $\mathcal{J}_{\tau,\delta}$ represents the cumulative displacement due to resampling. Recall that at each step $k$, the probability of a particle undergoing a resampling update is controlled by the jump intensity:

$$p_k = \Lambda(\mu_{t_k-}^N)\Delta t \leq K\Delta t.$$

The cumulative jump term $\mathcal{J}_{\tau,\delta}$ is non-zero only if at least one resampling event occurs in the interval $[\tau, \tau + \delta]$. Let $E_{\tau,\delta}$ be the event that at least one jump occurs during this interval. By the union bound

$$\mathbb{P}(E_{\tau,\delta}) \leq \sum_{k \in \mathcal{K}_{\tau,\delta}} p_k \leq \left\lceil \frac{\delta}{\Delta t} \right\rceil \overline{\Lambda}\Delta t \leq \overline{\Lambda}\delta,$$

the probability that the jump term magnitude exceeds $\varepsilon/3$ is bounded by the probability that a jump occurs at all:

$$\mathbb{P}(\|\mathcal{J}_{\tau,\delta}\| > \varepsilon/3) \leq \mathbb{P}(\mathcal{J}_{\tau,\delta} \neq 0) \leq \mathbb{P}(E_{\tau,\delta}) \leq \overline{\Lambda}\delta.$$

Taking the limit $\delta \downarrow 0$, this probability vanishes.

Combining the estimates for the three components, we have

$$\lim_{\delta \downarrow 0} \limsup_{\Delta t \to 0} \mathbb{P}\left(\|z_{\tau+\delta}^{\Delta t} - z_\tau^{\Delta t}\| > \varepsilon\right) = 0.$$

This verifies the Aldous criterion. Together with the pointwise tightness, the sequence $\{z_t^{\Delta t}\}$ is tight in $\mathcal{D}([0, T]; (\mathbb{R}^d \times \mathbb{R})^{\times N})$ and the proof is complete. $\qquad\square$

### A.4. Proof of Lemma A.4

We are finally ready to prove the main lemma that establishes the convergence of the discrete-time particle system to the continuous-time particle system.

*Proof of Lemma A.4.* With these lemmas, we conclude the convergence. Since (10) admits a unique strong solution, by an extended Yamada-Watanabe Theorem in (Barczy et al., 2015) we know that pathwise uniqueness indicates uniqueness in law. By Theorem 8.10, Chapter 4 in (Ethier and Kurtz, 1986), we conclude that $\{X^{(i),G,\Delta t}, w^{(i),\Delta t}\}_{i=1}^N$ converges weakly to $\{X^{(i),G}, w^{(i)}\}_{i=1}^N$ as $\Delta t \to 0$. $\qquad\square$

# B. Missing Proofs in Section 3

In this section, we provide the proofs of the theoretical results in Section 3. All proofs are based on assumptions in Appendix A.

## B.1. AFDPS / FK-Corrector Reweighting

We first derive the explicit form of the reweighting term used in the `AFDPS` dynamics (Chen et al., 2025).

`AFDPS` provides an alternative strategy for sampling from the posterior distribution $q(x) \propto p_{\text{data}}(x)e^{r(x)}$. In contrast to URGE, `AFDPS` can be interpreted as solving a high-dimensional PDE that governs posterior distribution evolution using the (stochastic) weighted particle method. For comparison with the notation of URGE, we set the parameters in their notations as $\boldsymbol{H}(x,t) = v(x,t)$ and $\mu_{\boldsymbol{y}} = -r$. The underlying unguided dynamics are defined by the canonical SDE

$$\mathrm{d}X_t = v(X_t,t)\mathrm{d}t + V(t)\mathrm{d}W_t,$$

on $[0,T]$, with $p_0 = \mathcal{N}(0,I_d)$ and terminal density $p_T = p_{\text{data}}$. Introducing guidance yields the particle evolution

$$\mathrm{d}X_t = \big(v(X_t,t) + V^2(t)\nabla_x r(X_t,t)\big)\mathrm{d}t + V(t)\mathrm{d}W_t,$$

where $r(x,t)$ denotes the reward.

The diffusion horizon is partitioned into intervals $0 = t_0 < t_1 < \cdots < t_K = T$. We initialize an $N$-particle system with $X_0^{(i)} \sim q_0(x) \propto p_0(x)e^{r(x,0)}$ and $\beta_0^{(i)} = 1$ for $i = 1, \ldots, N$. On each interval $[t_{k-1}, t_k)$, the particle states and weights evolve according to the Euler-Maruyama discretization of the guided dynamics. At grid points $t_k$ ($k = 1, \ldots, K-1$), we perform multinomial resampling with a prescribed threshold based on $\{X_{t_k}^{(i)}, \beta_{t_k}^{(i)}\}_{i=1}^N$:

$$\{\widehat{X}_{t_k}^{(i)}\}_{i=1}^N \sim \text{Categorical}\left(\{X_{t_k}^{(i)}\}_{i=1}^N, \left\{\frac{\beta_{t_k}^{(i)}}{\sum_{i=1}^N \beta_{t_k}^{(i)}}\right\}_{i=1}^N\right),$$

then set $X_{t_k}^{(i)} = \widehat{X}_{t_k}^{(i)}$ and reset $\beta_{t_k}^{(i)} = 1$. Here, $\beta_t^{(i)}$ denote the resampling weight of $i$-th particle at time $t$.

This decomposition of state and weight evolution follows from applying the weighted particle method to the unnormalized posterior $Q_t(x) = p_t(x)e^{r(x,t)}$. Its dynamics satisfy the Fokker-Planck equation (Chen et al., 2025, Lemma B.1):

$$\begin{aligned}
\frac{\partial}{\partial t}Q_t(x) = &-\nabla_x \cdot \Big[\big(v(x,t) + V^2(t)\nabla_x r(x,t)\big)Q_t(x)\Big] + \frac{1}{2}V^2(t)\Delta_x Q_t(x) \\
&+ \Big[\frac{1}{2}V^2(t)\big(\|\nabla_x r(x,t)\|^2 + \Delta_x r(x,t)\big) + \nabla_x r(x,t)^\top v(x,t) + \partial_t r(x,t)\Big]Q_t(x),
\end{aligned} \tag{12}$$

The reweighting term in (12) differs from that in the original equation by an additional $\partial_t r(x,t)$, which arises from the time dependence of the reward function $r$. Specifically, when $r$ depends on time, we have

$$\frac{\partial}{\partial t}p_t(x) = \frac{\partial}{\partial t}\big(Q_t(x)e^{-r(x,t)}\big) = e^{-r(x,t)}\big(\partial_t Q_t(x) - \partial_t r(x,t)Q_t(x)\big).$$

Consequently, the coefficient of $Q_t$ in the reweighted dynamics must include an additional $\partial_t r$ term.

The divergence term of (12) corresponds to the drift of the guided diffusion $X_t$, and the final term corresponds to the growth rate of the particle weight $\beta_t$. A detailed proof of this correspondence is provided in Lemmas B.2 and B.4 of (Chen et al., 2025).

To align the analysis with the guided state diffusion process using $G$, we replace the drift term $v(x,t) + V^2(t)\nabla_x r(x,t)$ with $v(x,t) + V^2(t)\nabla_x G(x,t)$. The corresponding modification in the weight dynamics is given by the following lemma.

**Lemma B.1.** *Let the particle state evolve according to the guided SDE*

$$\mathrm{d}X_t = \big(v(X_t, t) + V^2(t)\nabla_x G(X_t, t)\big)\mathrm{d}t + V(t)\mathrm{d}W_t,$$

*then the unnormalized posterior $Q_t = p_t e^{r(\cdot, t)}$ satisfies*

$$\frac{\partial}{\partial t}Q_t(x) = -\nabla_x \cdot \big((v(x,t) + V^2(t)\nabla_x G(x,t))Q_t(x)\big) + \frac{1}{2}V^2(t)\Delta_x Q_t(x)$$

$$+ \Big[\partial_t r(x,t) + \nabla_x r(x,t)^\top v(x,t) + V^2(t)\Big(-\frac{1}{2}\big(\|\nabla_x r(x,t)\|^2 + \Delta_x r(x,t)\big)$$

$$+ \Delta_x G(x,t) + \nabla_x \log p_t(x)^\top \nabla_x (G-r)(x,t) + \nabla_x G(x,t)^\top \nabla_x r(x,t)\Big)\Big]Q_t(x).$$

*Proof.* By Lemmas B.2 and B.4 in (Chen et al., 2025), the weight dynamics follow from the Fokker-Planck equation for $Q_t$, with the drift replaced by $v(x,t) + V^2(t)\nabla_x G(x,t)$.

A direct computation yields

$$\frac{\partial}{\partial t}Q_t(x) = -\nabla_x \cdot \big((v(x,t) + V^2(t)\nabla_x G(x,t))Q_t(x)\big) + V^2(t)\nabla_x \cdot \big((\nabla_x G(x,t) - \nabla_x r(x,t))Q_t(x)\big) + \frac{1}{2}V^2(t)\Delta_x Q_t(x)$$

$$+ \Big(\frac{1}{2}V^2(t)(\|\nabla_x r(x,t)\|^2 + \Delta_x r(x,t)) + (\nabla_x r(x,t))^\top v(x,t) + \partial_t r(x,t)\Big)Q_t(x)$$

$$= -\nabla_x \cdot \big((v(x,t) + V^2(t)\nabla_x G(x,t))Q_t(x)\big) + V^2(t)(\nabla_x G(x,t) - \nabla_x r(x,t))^\top \nabla_x Q_t(x)$$

$$+ V^2(t)(\Delta_x G(x,t) - \Delta_x r(x,t))Q_t(x) + \frac{1}{2}V^2(t)\Delta_x Q_t(x)$$

$$+ \Big(\frac{1}{2}V^2(t)(\|\nabla_x r(x,t)\|^2 + \Delta_x r(x,t)) + (\nabla_x r(x,t))^\top v(x,t) + \partial_t r(x,t)\Big)Q_t(x)$$

$$= -\nabla_x \cdot \big((v(x,t) + V^2(t)\nabla_x G(x,t))Q_t(x)\big)$$

$$+ V^2(t)(\nabla_x G(x,t) - \nabla_x r(x,t))^\top (\nabla_x \log p_t(x) + \nabla_x r(x,t))Q_t(x)$$

$$+ V^2(t)(\Delta_x G(x,t) - \Delta_x r(x,t))Q_t(x) + \frac{1}{2}V^2(t)\Delta_x Q_t(x)$$

$$+ \Big(\frac{1}{2}V^2(t)(\|\nabla_x r(x,t)\|^2 + \Delta_x r(x,t)) + (\nabla_x r(x,t))^\top v(x,t) + \partial_t r(x,t)\Big)Q_t(x),$$

where we used

$$\nabla_x Q_t(x) = \nabla_x\big(p_t(x)e^{r(x,t)}\big) = e^r(x,t)\nabla_x p_t(x) + p_t(x)e^r(x,t)\nabla_x r(x,t) = Q_t(x)(\nabla_x \log p_t(x) + \nabla_x r(x,t)).$$

in the last equation. Collecting terms gives the claimed expression. $\square$

Consequently, choosing $G = r$ recovers the `AFDPS` drift and weight update, while $G = 0$ yields the `FK-Corrector` dynamics.

### B.2. Continuous approximation-freeness

For readers' convenience, we reiterate Theorem 3.1 here.

**Theorem B.2** (Continuous approximation-freeness). *For any test function $\varphi \in C_b^{2,1}(\mathbb{R}^d \times \mathbb{R})$, if the initialization at time $s$ satisfies $\mathbb{E}_{\overline{\mathbb{P}}}[M_s^N(\varphi)] = \mathbb{E}_{\mathbb{P}}[\varphi(X_s^G, s)]$, then at terminal time $T$, we have*

$$\mathbb{E}_{\overline{\mathbb{P}}}[M_t^N(\varphi)] = \frac{\mathbb{E}_{\mathbb{P}}[e^{r(X_t^G, t) - r(X_s^G, s)}\varphi(X_t^G, t)]}{\mathbb{E}_{\mathbb{P}}[e^{r(X_t, t) - r(X_s, s)}]}.$$

*Proof of Theorem 3.1.* The proof consists of two parts. First, the empirical distribution of the weighted particles generated by URGE converges to that of its continuous-time limit, as established in Lemma A.4. Second, the expectation of $M_t^N(\varphi)$ under the continuous-time formulation is approximation-free. In what follows, we verify the correctness of the second part.

Fix $s$. We study the infinitesimal evolution of $M_t^N(\varphi)$. By Appendix A, since $\Delta t \to 0$, our analysis can be transferred to $\sum_{i=1}^N \frac{w_t^{(i)}}{\sum_{j=1}^N w_t^{(j)}} \varphi(X_t^{(i),G}, t)$, following the symbols in Appendix A and without abuse of notation, we also call this term $M_t^N(\varphi)$. Denote $W_t = \sum_{j=1}^N w_t^{(j)}$, and $\overline{w}_t^{(i)} := \frac{w_t^{(i)}}{W_t}$.

By Itô's formula,

$$d\left(w_t^{(i)}\varphi(X_t^{(i),G}, t)\right) = w_t^{(i)} d\varphi(X_t^{(i),G}, t) + dw_t^{(i)}\varphi(X_t^{(i),G}, t) + d\langle w^{(i)}, \varphi(X^{(i),G}, t)\rangle_t.$$

The generator of base measure is

$$(\mathcal{L}\varphi)(x) = \partial_t\varphi(x, t) + v(x, t)^\top \nabla_x\varphi(x, t) + \frac{1}{2}V^2(t)\Delta_x\varphi(x, t).$$

Between resampling times, particles follow the guided SDE and their log-weights evolve as

$$d\log w_t^{(i)} = dr(X_t^{(i),G}, t) - (\theta_t^{(i),G})^\top dW_t^{(i)} - \frac{1}{2}\|\theta_t^{(i),G}\|^2 dt,$$

where $\theta_t^{(i),G} = V(t)\nabla_x G(X_t^{(i),G}, t)$. Since

$$dr(X_t^{(i),G}, t) = \left((\mathcal{L}r)(X_t^{(i),G}, t) + V^2(t)\nabla_x G(X_t^{(i),G}, t)^\top \nabla_x r(X_t^{(i),G}, t)\right)dt + V(t)\nabla_x r(X_t^{(i),G}, t)^\top dW_t^{(i)},$$

by applying Itô's formula to $w = \exp(\log w)$ gives

$$dw_t^{(i)} = w_t^{(i)}(V(t)\nabla_x r(X_t^{(i),G}, t) - \theta_t^{(i),G})^\top dW_t^{(i)} + w_t^{(i)}\left((\mathcal{L}r)(X_t^{(i),G}, t) + \frac{1}{2}V^2(t)\|\nabla_x r(X_t^{(i),G}, t)\|^2\right)dt.$$

Denote $a_t^{(i)} := (\mathcal{L}r)(X_t^{(i),G}, t) + \frac{1}{2}V^2(t)\|\nabla_x r(X_t^{(i),G}, t)\|^2$, and $\sigma_t^{(i)} := V(t)\nabla_x r(X_t^{(i),G}, t) - \theta_t^{(i),G}$.

The differential of the total weight $W_t$ is given by

$$dW_t = \sum_{j=1}^N dw_t^{(j)} = \sum_{j=1}^N w_t^{(j)}\sigma_t^{(j)\top} dW_t^{(j)} + \sum_{j=1}^N w_t^{(j)} a_t^{(j)} dt.$$

By applying Itô's formula to the function $f(x) = x^{-1}$ yields

$$d\left(\frac{1}{W_t}\right) = -\frac{1}{W_t^2} dW_t + \frac{1}{W_t^3} d\langle W\rangle_t,$$

where the quadratic variation of $W_t$ is given by $d\langle W\rangle_t = \sum_{j=1}^N (w_t^{(j)})^2\|\sigma_t^{(j)}\|^2 dt$ using the independence of the Brownian motions. Substituting $dW_t$ and $d\langle W\rangle_t$ into the above expression, we obtain

$$d\left(\frac{1}{W_t}\right) = -\frac{1}{W_t^2}\sum_{j=1}^N w_t^{(j)}\sigma_t^{(j)\top} dW_t^{(j)} - \frac{1}{W_t^2}\sum_{j=1}^N w_t^{(j)} a_t^{(j)} dt + \frac{1}{W_t^3}\sum_{j=1}^N (w_t^{(j)})^2\|\sigma_t^{(j)}\|^2 dt.$$

We now apply the product rule to $\overline{w}_t^{(i)} = w_t^{(i)} W_t^{-1}$:

$$d\overline{w}_t^{(i)} = \frac{1}{W_t} dw_t^{(i)} + w_t^{(i)} d\left(\frac{1}{W_t}\right) + d\left\langle w^{(i)}, \frac{1}{W}\right\rangle_t.$$

The first term reads

$$\frac{1}{W_t} dw_t^{(i)} = \overline{w}_t^{(i)}\sigma_t^{(i)\top} dW_t^{(i)} + \overline{w}_t^{(i)} a_t^{(i)} dt.$$

For the second term, using $w_t^{(i)}/W_t = \overline{w}_t^{(i)}$, we obtain

$$w_t^{(i)} \, \mathrm{d}\Big(\frac{1}{W_t}\Big) = -\overline{w}_t^{(i)} \sum_{j=1}^{N} \overline{w}_t^{(j)} \sigma_t^{(j)\top} \mathrm{d}W_t^{(j)} - \overline{w}_t^{(i)} \sum_{j=1}^{N} \overline{w}_t^{(j)} a_t^{(j)} \, \mathrm{d}t + \overline{w}_t^{(i)} \sum_{j=1}^{N} (\overline{w}_t^{(j)})^2 \|\sigma_t^{(j)}\|^2 \, \mathrm{d}t.$$

Finally, the cross-variation term only receives contribution from the common Brownian motion $W^{(i)}$:

$$\mathrm{d}\Big\langle w^{(i)}, \frac{1}{W} \Big\rangle_t = -\frac{(w_t^{(i)})^2}{W_t^2} \|\sigma_t^{(i)}\|^2 \, \mathrm{d}t = -(\overline{w}_t^{(i)})^2 \|\sigma_t^{(i)}\|^2 \, \mathrm{d}t.$$

Collecting all terms, we arrive at

$$\mathrm{d}\overline{w}_t^{(i)} = \overline{w}_t^{(i)} \Big(\sigma_t^{(i)\top} \mathrm{d}W_t^{(i)} - \sum_{j=1}^{N} \overline{w}_t^{(j)} \sigma_t^{(j)\top} \mathrm{d}W_t^{(j)}\Big) + \overline{w}_t^{(i)} \Big(a_t^{(i)} - \sum_{j=1}^{N} \overline{w}_t^{(j)} a_t^{(j)}\Big) \, \mathrm{d}t + \overline{w}_t^{(i)} \sum_{j=1}^{N} (\overline{w}_t^{(j)})^2 \|\sigma_t^{(j)}\|^2 \, \mathrm{d}t - (\overline{w}_t^{(i)})^2 \|\sigma_t^{(i)}\|^2 \, \mathrm{d}t.$$

Then we come back to the analysis of $M_t^N$. The guided particles satisfy

$$\mathrm{d}\varphi\big(X_t^{(i),G}, t\big) = \big(\mathcal{L}_t^G \varphi\big)\big(X_t^{(i),G}, t\big) \, \mathrm{d}t + V(t)\nabla\varphi\big(X_t^{(i),G}, t\big)^\top \mathrm{d}W_t^{(i)},$$

with $\mathcal{L}_t^G = \mathcal{L}_t + V^2(t)\nabla G^\top \nabla$.

Applying Itô's product rule to $\overline{w}_t^{(i)} \varphi\big(X_t^{(i),G}, t\big)$ and summing over $i = 1, \dots, N$, we obtain

$$\mathrm{d}M_t^N(\varphi) = \sum_{i=1}^{N} \varphi\big(X_t^{(i),G}, t\big) \, \mathrm{d}\overline{w}_t^{(i)} + \sum_{i=1}^{N} \overline{w}_t^{(i)} \, \mathrm{d}\varphi\big(X_t^{(i),G}, t\big) + \sum_{i=1}^{N} \mathrm{d}\Big\langle \overline{w}^{(i)}, \varphi\big(X^{(i),G}, t\big) \Big\rangle_t.$$

The drift terms can be grouped as follows. First,

$$\sum_{i=1}^{N} \overline{w}_t^{(i)} \, \mathrm{d}\varphi\big(X_t^{(i),G}, t\big) = M_t^N(\mathcal{L}\varphi) \, \mathrm{d}t + M_t^N\big(V^2 \nabla G^\top \nabla \varphi\big) \, \mathrm{d}t + \mathrm{d}M_{1,t}.$$

where $M_{1,t}$ is a local martingale. Second,

$$\sum_{i=1}^{N} \varphi\big(X_t^{(i),G}, t\big) \, \mathrm{d}\overline{w}_t^{(i)} = \sum_{i=1}^{N} \varphi\big(X_t^{(i),G}, t\big) \, \overline{w}_t^{(i)} \Big(a_t^{(i)} - \sum_{j=1}^{N} \overline{w}_t^{(j)} a_t^{(j)}\Big) \mathrm{d}t + \sum_{i=1}^{N} \varphi\big(X_t^{(i),G}, t\big) \, \overline{w}_t^{(i)} \sum_{j=1}^{N} \big(\overline{w}_t^{(j)}\big)^2 \|\sigma_t^{(i)}\|^2 \mathrm{d}t$$

$$+ \sum_{i=1}^{N} \varphi\big(X_t^{(i),G}, t\big) \big(\overline{w}_t^{(i)}\big)^2 \|\sigma_t^{(i)}\|^2 \mathrm{d}t. \tag{13}$$

The last two terms can be bounded by

$$\Big\| \sum_{i=1}^{N} \varphi\big(X_t^{(i),G}, t\big) \, \overline{w}_t^{(i)} \sum_{j=1}^{N} \big(\overline{w}_t^{(j)}\big)^2 \|\sigma_t^{(i)}\|^2 + \sum_{i=1}^{N} \varphi\big(X_t^{(i),G}, t\big) \big(\overline{w}_t^{(i)}\big)^2 \|\sigma_t^{(i)}\|^2 \Big\|_\infty \leq C \sum_{i=1}^{N} \big(\overline{w}_t^{(i)}\big)^2,$$

where $C$ is a constant depending on uniform bounds of $r, G, V$, as well as the parabolic $W_\infty^{2,1}$ norm $\|\varphi\|_{W_\infty^{2,1}}$, defined by $\|f\|_{W_\infty^{2,1}} := \|f\|_\infty + \|\partial_t f\|_\infty + \|\nabla_x f\|_\infty + \|\nabla_x^2 f\|_\infty$.

For each propagation interval $[t_1, t_2)$, where $t_1$ and $t_2$ are two consecutive jump times, we have $|w_t^{(i)} - w_{t_1}^{(i)}| \leq C(t_2 - t_1)$, for some constant $C$ depending only on uniform bounds of $v, r, G, V$, independently of time and particle index. Moreover, $w_{t_1}^{(i)} \equiv w_{t_1}$ for all $i$. Let $\Delta t = t_2 - t_1$. As $\Delta t \to 0$,

$$\sum_{i=1}^{N} (\overline{w}_t^{(i)})^2 = \sum_{i=1}^{N} \Big(\frac{w_t^{(i)}}{\sum_{j=1}^{N} w_t^{(j)}}\Big)^2 \leq \sum_{i=1}^{N} \Big(\frac{w_{t_1} + C(t - t_1)}{N w_{t_1} - NC(t - t_1)}\Big)^2 \leq \sum_{i=1}^{N} \Big(\frac{\frac{3}{2} w_{t_1}}{N \frac{1}{2} w_{t_1}}\Big)^2 = \frac{9}{N}.$$

Thus, (13) can be further simplified as

$$\sum_{i=1}^{N} \varphi\big(X_t^{(i),G}, t\big)\, \mathrm{d}\overline{w}_t^{(i)} = M_t^N(\varphi a_t)\, \mathrm{d}t - M_t^N(\varphi)\, M_t^N(a_t)\, \mathrm{d}t + O(\frac{1}{N}).$$

where big-$O$ only depends on the coefficients of the jump-diffusion system.

Third, the quadratic covariation between $\overline{w}_t^{(i)}$ and $\varphi(X_t^{(i),G})$ gives

$$\sum_{i=1}^{N} \mathrm{d}\big\langle \overline{w}^{(i)}, \varphi\big(X^{(i),G}\big)\big\rangle_t = \sum_{i=1}^{N} V(t)\nabla\varphi^\top \sigma_t^{(i)}(\overline{w}_t^{(i)} - (\overline{w}_t^{(i)})^2)\mathrm{d}t$$

$$= M_t^N\big(V^2\nabla r^\top \nabla\varphi\big)\, \mathrm{d}t - M_t^N\big(V^2\nabla G^\top \nabla\varphi\big)\, \mathrm{d}t + O(\frac{1}{N}),$$

The $O(\frac{1}{N})$ term is bounded similarly as previous discussion on $\big(\overline{w}_t^{(i)}\big)^2$.

Collecting all contributions, we arrive at

$$\mathrm{d}M_t^N(\varphi) = M_t^N(\mathcal{L}\varphi)\, \mathrm{d}t + M_t^N\big(V^2\nabla r^\top \nabla\varphi\big)\, \mathrm{d}t + M_t^N(\varphi a_t)\, \mathrm{d}t - M_t^N(\varphi)\, M_t^N(a_t)\, \mathrm{d}t + \mathrm{d}\mathcal{M}_t^N + O(\frac{1}{N}), \qquad (14)$$

where $\mathcal{M}_t^N$ is a local martingale collecting all terms proportional to $\mathrm{d}W_t^{(i)}$.

Let $\tau$ be a jump time of the Poissonization resampling process. Denote pre- and post-jump states by $(X_{\tau-}^G, w_{\tau-})$ and $(X_\tau^G, w_\tau)$. Weights become uniform as $w_\tau^{(i)} = \frac{1}{N}\sum_{k=1}^{N} w_{\tau-}^{(k)}$. Each $X_\tau^{(i),G}$ is drawn independently from

$$\sum_{j=1}^{N} \frac{w_{\tau-}^{(j)}}{\sum_{k=1}^{N} w_{\tau-}^{(k)}} \delta_{X_{\tau-}^{(j),G}}.$$

Hence,

$$\mathbb{E}_{\overline{\mathbb{P}}}[M_\tau^N(\varphi) \mid \mathcal{F}_{\tau-}] = \sum_{i=1}^{N} \mathbb{E}_{\overline{\mathbb{P}}}\big[\overline{w}_\tau^{(i)}\varphi(X_\tau^{(i),G}) \mid \mathcal{F}_{\tau-}\big] = \mathbb{E}_{\overline{\mathbb{P}}}[\varphi(X_\tau^{(1),G}) \mid \mathcal{F}_{\tau-}] = \sum_{j=1}^{N} \frac{w_{\tau-}^{(j)}}{\sum_{k=1}^{N} w_{\tau-}^{(k)}}\varphi(X_{\tau-}^{(j),G}) = M_{\tau-}^N(\varphi).$$

where $\mathcal{F}_{\tau-}$ denotes information known at time $\tau-$. Thus jumps add no drift to the expectation.

Define $m_t(\varphi) := \mathbb{E}_{\overline{\mathbb{P}}}[M_t^N(\varphi)]$. Taking expectations in (14) and applying the jump calculation gives

$$\frac{\mathrm{d}}{\mathrm{d}t}m_t(\varphi) = m_t\big(\mathcal{L}\varphi + V^2(t)\nabla r^\top \nabla\varphi + \varphi a_t\big) - \mathbb{E}_{\overline{\mathbb{P}}}[M_t^N(\varphi)M_t^N(a_t)] + O(\frac{1}{N}). \qquad (15)$$

Further define

$$A_N := \frac{1}{N}\sum_{i=1}^{N} w_t^{(i)}\varphi(X_t^{(i),G}, t), \quad B_N := \frac{1}{N}\sum_{i=1}^{N} w_t^{(i)}a_t^{(i)}, \quad C_N := \frac{1}{N}\sum_{i=1}^{N} w_t^{(i)}.$$

Then $M_t^N(\varphi) = \frac{A_N}{C_N}$, $M_t^N(a_t) = \frac{B_N}{C_N}$. Assume that

$$\mathrm{Var}(A_N) = O\Big(\frac{1}{N}\Big), \quad \mathrm{Var}(B_N) = O\Big(\frac{1}{N}\Big), \quad \mathrm{Var}(C_N) = O\Big(\frac{1}{N}\Big).$$

Let $\mu_A = \mathbb{E}[A_N]$, $\mu_B = \mathbb{E}[B_N]$, and $\mu_C = \mathbb{E}[C_N]$. We write $A_N = \mu_A + \delta_A$, $B_N = \mu_B + \delta_B$, $C_N = \mu_C + \delta_C$, where $\mathbb{E}[\delta_.] = 0$ and $\mathbb{E}[\delta^2] = O(1/N)$. Using a first-order Taylor expansion,

$$\frac{1}{C_N} = \frac{1}{\mu_C + \delta_C} = \frac{1}{\mu_C}\Big(1 - \frac{\delta_C}{\mu_C}\Big) + O\Big(\frac{1}{N}\Big).$$

Therefore,

$$M_t^N(\varphi) = \frac{A_N}{C_N} = \frac{\mu_A}{\mu_C} + \frac{\delta_A}{\mu_C} - \frac{\mu_A \delta_C}{\mu_C^2} + O\left(\frac{1}{N}\right),$$

and similarly,

$$M_t^N(a_t) = \frac{\mu_B}{\mu_C} + \frac{\delta_B}{\mu_C} - \frac{\mu_B \delta_C}{\mu_C^2} + O\left(\frac{1}{N}\right).$$

Since $\delta_A, \delta_B, \delta_C$ all have variance of order $O(1/N)$, it follows that

$$\text{Cov}\big(M_t^N(\varphi), M_t^N(a_t)\big) = O\left(\frac{1}{N}\right).$$

(15) can be further simplified into

$$\frac{\mathrm{d}}{\mathrm{d}t} m_t(\varphi) = m_t\left(\mathcal{L}\varphi + V^2(t)\nabla r^\top \nabla\varphi + \varphi a_t\right) - m_t(\varphi)m_t(a_t) + O(\frac{1}{\sqrt{N}}).$$

To verify the explicit representation, define a candidate measure $\widetilde{m}_t$ by its action on test functions:

$$\widetilde{m}_t(\varphi) := \frac{\mathbb{E}_{\mathbb{P}}\left[e^{r(X_t,t) - r(X_s,s)}\varphi(X_t,t)\right]}{\mathbb{E}_{\mathbb{P}}\left[e^{r(X_t,t) - r(X_s,s)}\right]}.$$

We compute the time derivative of $\widetilde{m}_t(\varphi)$.

Define $\gamma_t(\varphi) := \mathbb{E}_{\mathbb{P}}\left[e^{r(X_t,t) - r(X_s,s)}\varphi(X_t,t)\right]$. By direct computation, we obtain

$$\frac{\mathrm{d}}{\mathrm{d}t}\gamma_t(\varphi) = \gamma_t(\mathcal{L}\varphi + V^2\nabla r^\top \nabla\varphi + \varphi a_t).$$

Thus, we have

$$\frac{\mathrm{d}}{\mathrm{d}t}\widetilde{m}_t(\varphi) = \frac{\mathrm{d}}{\mathrm{d}t}\left(\frac{\gamma_t(\varphi)}{\gamma_t(1)}\right) = \frac{\gamma_t(\mathcal{L}\varphi + V^2\nabla r^\top \nabla\varphi + \varphi a_t)}{\gamma_t(1)} - \frac{\gamma_t(\varphi)\left(\frac{\mathrm{d}}{\mathrm{d}t}\gamma_t(1)\right)}{(\gamma_t(1))^2} = \widetilde{m}_t(\mathcal{L}\varphi + V^2\nabla r^\top \nabla\varphi + \varphi a_t) - \widetilde{m}_t(\varphi)\widetilde{m}_t(a_t).$$

By taking $N \to \infty$, $m_t(\varphi)$ and $\widetilde{m}_t(\varphi)$ satisfy the same linear evolution equation with the same initial condition $m_0(\varphi) = \widetilde{m}_0(\varphi) = \mathbb{E}_{\mathbb{P}}[\varphi(X_s)]$, they must be identical. Thus, $\mathbb{E}_{\overline{\mathbb{P}}}[M_t^N(\varphi)] = \frac{\mathbb{E}_{\mathbb{P}}\left[e^{r(X_t,t) - r(X_s,s)}\varphi(X_t,t)\right]}{\mathbb{E}_{\mathbb{P}}\left[e^{r(X_t,t) - r(X_s,s)}\right]}$, and the proof is complete. $\qquad\square$

## B.3. Effective Marginalized Generator

For readers' convenience, we reiterate Theorem 3.2 here.

**Theorem B.3** (Effective Marginalized Generator). *Define infinitesimal generator $\mathcal{L}_t^G$ for $\varphi \in C_b^{2,1}(\mathbb{R}^d \times \mathbb{R})$ by*

$$\mathcal{L}_t^G\varphi(x,t) = \partial_t\varphi(x,t) + \frac{1}{2}\text{tr}\big(V^2(t)\nabla^2\varphi(x,t)\big) + \big(v(x,t) + V^2(t)\nabla_x G(x,t)\big)^\top \nabla\varphi(x,t).$$

*Assume that for each $(x,t)$, the instantaneous intensity of the weight exists and is given by*

$$\lambda(x,t) := \lim_{h\downarrow 0}\frac{1}{h}\left(\mathbb{E}_{\mathbb{P}^G}\left[w_{t-h,t}^{URGE}(X_{t-h:t}^{(i),G})\Big| X_t^{(i),G} = x\right] - 1\right).$$

*The value function $\psi(x,t)$ satisfies the generalized Kolmogorov backward equation:*

$$\partial_t\psi(x,t) + \mathcal{L}_t^G\psi(x,t) + (\lambda(x,t) - \bar{\lambda}_t)\psi(x,t) = 0.$$

*Proof of Theorem 3.2.* We consider the whole system as the continuous-time system mentioned in Appendix A, with $\tilde{\mathbb{P}}$ to be the law of the evolution. We first analyze the unnormalized value function.

Define $\Psi(x, t)$ as the expected unnormalized weighted contribution:

$$\Psi(x, t) := \mathbb{E}_{\tilde{\mathbb{P}}}\left[ \frac{1}{N} \sum_{i=1}^{N} w_{t,T}(X_{t:T}^{(i),G})\, \varphi(X_T^{(i),G}, T) \ \middle| \ X_t^{(i),G} = x \right]. \tag{16}$$

Fix $h > 0$ and decompose the time interval into $(t, t+h]$ and $(t+h, T]$. By the multiplicative property of the weights, $w_{t,T} = w_{t,t+h}\, w_{t+h,T}$, and the tower property, conditioning on the filtration $\mathcal{F}_{t+h}$ yields

$$\Psi(x, t) = \mathbb{E}_{\tilde{\mathbb{P}}}\left[ \mathbb{E}_{\tilde{\mathbb{P}}}\left[ \frac{1}{N} \sum_{j=1}^{N} w_{t,t+h}(X_{t:t+h}^{(j),G})\, w_{t+h,T}(X_{t+h:T}^{(j),G})\, \varphi(X_T^{(j),G}, T) \ \middle| \ \mathcal{F}_{t+h} \right] \ \middle| \ X_t^{(i),G} = x \right].$$

At time $t + h$, index the particles by $k = 1, \ldots, N$. Each particle $j$ at time $T$ is a descendant of a unique ancestor $k$ at time $t + h$. By the Markov property and exchangeability of the particle system, the inner conditional expectation can be grouped by ancestors, giving

$$\mathbb{E}_{\tilde{\mathbb{P}}}\left[ \frac{1}{N} \sum_{j=1}^{N} w_{t,t+h}(X_{t:t+h}^{(j),G})\, w_{t+h,T}(X_{t+h:T}^{(j),G})\, \varphi(X_T^{(j),G}, T) \ \middle| \ \mathcal{F}_{t+h} \right] = \frac{1}{N} \sum_{k=1}^{N} w_{t,t+h}^{(k)}\, \Psi(X_{t+h}^{(k),G}, t+h).$$

Therefore,

$$\Psi(x, t) = \mathbb{E}_{\tilde{\mathbb{P}}}\left[ \frac{1}{N} \sum_{k=1}^{N} w_{t,t+h}^{(k)}(X_{t:t+h}^{(k),G})\, \Psi(X_{t+h}^{(k),G}, t+h) \ \middle| \ X_t^{(i),G} = x \right]. \tag{17}$$

By approximation-freeness of the resampling mechanism (see discussion in Proof of Theorem 3.1), the jump generator annihilates linear test functions in expectation, and thus does not contribute to the first-order expansion. Consequently, we only need to consider the continuous evolution.

Conditioning on $X_t^{(i),G} = x$, contributions from particles $k \neq i$ are independent of $x$ and vanish when deriving the generator. Hence, it suffices to analyze a single particle trajectory. The weight increment admits the expansion

$$w_{t,t+h} = 1 + h\, \lambda(x, t) + o(h),$$

while the state evolves according to the generator $\mathcal{L}$. Applying Itô's formula yields

$$\mathbb{E}_{\tilde{\mathbb{P}}}\left[ w_{t,t+h}(X_{t:t+h}^G)\, \Psi(X_{t+h}^G, t+h) \mid X_t^G = x \right] = \Psi(x, t) + h\big(\partial_t \Psi(x, t) + \mathcal{L}\Psi(x, t) + \lambda(x, t)\Psi(x, t)\big) + o(h).$$

Substituting this expansion into (17), dividing by $h$, and letting $h \to 0$ yields the backward equation

$$\partial_t \Psi(x, t) + \mathcal{L}\Psi(x, t) + \lambda(x, t)\Psi(x, t) = 0.$$

We now derive the backward equation satisfied by the normalized estimator in the mean-field limit. Define

$$\psi(x, t) := \lim_{N \to \infty} \mathbb{E}\left[ \frac{\sum_{i=1}^{N} w_{t,T}(X_{t:t+h}^{(i),G})\, \varphi(X_T^{(i),G}, T)}{\sum_{j=1}^{N} w_{t,T}(X_{t:t+h}^{(j),G})} \ \middle| \ X_t^{(i),G} = x \right],$$

where the limit is understood under standard propagation-of-chaos assumptions for the interacting particle system. In particular, the empirical measure converges to a deterministic marginal law $\mu_t$, and both numerator and denominator satisfy a law of large numbers.

As a consequence, the denominator converges in probability to a deterministic normalizing factor, which represents the expected mass growth of the unnormalized system from time $t$ to $T$. We define

$$Z(t, T) := \exp\left( \int_t^T \bar{\lambda}_s\, ds \right), \qquad \bar{\lambda}_s := \mathbb{E}_{\tilde{\mathbb{P}}}\big[\lambda(X_s^G, s)\big],$$

where the expectation is taken with respect to the limiting marginal law $\mu_s$. The normalized value function is therefore related to the unnormalized one by

$$\psi(x,t) = \frac{\Psi(x,t)}{Z(t,T)}. \tag{18}$$

We first compute the time derivative of the normalizing factor. By the fundamental theorem of calculus,

$$\partial_t Z(t,T) = -\bar{\lambda}_t \, Z(t,T). \tag{19}$$

Differentiating (18) with respect to $t$ yields

$$\partial_t \psi(x,t) = \frac{(\partial_t \Psi(x,t)) \, Z(t,T) - \Psi(x,t) \, (\partial_t Z(t,T))}{Z(t,T)^2}.$$

Substituting the linear backward equation $\partial_t \Psi = -(\mathcal{L}\Psi + \lambda\Psi)$ and (19) gives

$$\partial_t \psi(x,t) = \frac{-\big(\mathcal{L}\Psi(x,t) + \lambda(x,t)\Psi(x,t)\big)Z(t,T) + \bar{\lambda}_t \, \Psi(x,t)\, Z(t,T)}{Z(t,T)^2}$$

$$= -\Big(\mathcal{L}\frac{\Psi(x,t)}{Z(t,T)} + (\lambda(x,t) - \bar{\lambda}_t)\frac{\Psi(x,t)}{Z(t,T)}\Big).$$

Using (18), we obtain the nonlinear backward equation $\partial_t \psi(x,t) + \mathcal{L}\psi(x,t) + \big(\lambda(x,t) - \bar{\lambda}_t\big)\psi(x,t) = 0$.

$\square$

## B.4. Marginalized Equivalence between `URGE` and `FK-Corrector/AFDPS`

For readers' convenience, we reiterate Theorem 3.3 here.

**Theorem B.4** (Marginalized Equivalence between `URGE` and `FK-Corrector/AFDPS`). *The convergence rate of* `URGE` *to* `FK-Corrector/AFDPS` *is given by*

$$\lim_{s\to t} \frac{\mathbb{E}_{\mathbb{P}^G}\big[w_{URGE}\big|X_t^G = x\big] - 1}{t - s} = w_{AFDPS}(x),$$

*where the pathwise weight of* `URGE` *is defined as*

$$w_{URGE} = \exp\left(\int_s^t -V(\tau)\nabla_x G(X_\tau^G, \tau)^\top \mathrm{d}W_\tau - \frac{1}{2}V^2(\tau)\|\nabla_x G(X_\tau^G,\tau)\|_2^2 \mathrm{d}\tau + \int_s^t \mathrm{d}r(X_\tau^G,\tau)\right).$$

*and the pathwise weight of* `FK-Corrector/AFDPS` *is given by*

$$w_{AFDPS}(x) = \partial_t r(x,t) - \tfrac{1}{2}V^2(t)\big(\Delta_x r(x,t) + \|\nabla_x r(x,t)\|_2^2\big)$$
$$+ \nabla_x r(x,t)^\top \big(v(x,t) + V^2(t)\nabla_x G(x,t)\big)$$
$$+ V^2(t)\nabla_x \log p_t(x)^\top \nabla_x (G - r)(x,t) + V^2(t)\Delta_x G(x,t).$$

*Proof of Theorem 3.3.* We first simplify the left-hand side of (8). By the assumption on $r$ and Itô's formula,

$$\mathrm{d}r(X_t^G, t) = \partial_t r(X_t^G, t)\mathrm{d}t + \nabla_x r(X_t^G, t)^\top \big(v(X_t^G, t) + V^2(t)\nabla_x G(X_t^G, t)\big)\mathrm{d}t$$
$$+ \frac{1}{2}V^2(t)\Delta_x r(X_t^G, t)\mathrm{d}t + V(t)\nabla_x r(X_t^G, t)^\top \mathrm{d}W_t.$$

Hence,

$$\lim_{s\to t} \frac{\mathbb{E}_{\mathbb{P}^G}\left[\exp\left(\int_s^t \big(-V(\tau)\nabla_x G(X_\tau^G, \tau)^\top \mathrm{d}W_\tau - \frac{1}{2}V^2(\tau)\|\nabla_x G(X_\tau^G,\tau)\|_2^2 \mathrm{d}\tau + \mathrm{d}r(X_\tau^G,\tau)\big)\right)\Big|X_t^G = x\right] - 1}{t - s}$$

$$= \lim_{s \to t} \frac{1}{t-s} \left( \mathbb{E}_{\mathbb{P}^G} \left[ \exp \left( \int_s^t V(\tau) \nabla_x (r-G)(X_\tau^G, \tau)^\top \mathrm{d}W_\tau \right. \right. \right.$$

$$+ \left( \partial_t r(X_\tau^G, \tau) - \frac{1}{2} V^2(\tau) \| \nabla_x G(X_\tau^G, \tau) \|_2^2 + \nabla_x r(X_\tau^G, \tau)^\top \left( v(X_\tau^G, \tau) + V^2(\tau) \nabla_x G(X_\tau^G, \tau) \right) \right.$$

$$\left. \left. \left. + \frac{1}{2} V^2(\tau) \Delta_x r(X_\tau^G, \tau) \right) \mathrm{d}s \right) \Big| X_T^G = x \right] - 1 \right). \tag{20}$$

Because $p_t$ is smooth and satisfies the Fokker-Planck equation, and since the terminal law of $X_t$ is $Q_t \propto p_t e^{r(\cdot, T)}$, the time-reversal formula yields a backward Brownian motion $\overline{W}_\tau$ such that

$$\mathrm{d}W_\tau = \mathrm{d}\overline{W}_\tau - V(\tau) \nabla_x \log Q_\tau(X_\tau^G) \mathrm{d}\tau.$$

Thus,

$$\int_s^t V(\tau) \nabla_x (r-G)(X_\tau^G, \tau)^\top \mathrm{d}W_\tau = \int_t^T V(\tau) \nabla_x (r-G)(X_\tau^G, \tau)^\top \mathrm{d}\overline{W}_\tau \tag{21}$$

$$- \int_t^T V^2(\tau) \nabla_x (r-G)(X_\tau^G, \tau)^\top \nabla_x \log Q_\tau(X_\tau^G) \mathrm{d}\tau, \tag{22}$$

where $Q_s$ denotes the law of $X_s^G$ under the terminal condition $Q_T \propto p_T e^{r(\cdot, T)}$.

We next rewrite the forward stochastic integral with respect to $\mathrm{d}\overline{W}_\tau$ as a backward Itô integral. Let $H_\tau := V(\tau) \nabla_x (r - G)(X_\tau^G, \tau)$. Under the associated measure transformation,

$$\mathrm{d}H_\tau = A_\tau \mathrm{d}\tau + B_\tau \mathrm{d}\overline{W}_\tau, \quad B_\tau = V^2(\tau) \nabla_x^2 (r-G)(X_\tau^G, \tau).$$

For processes of this form, Haussmann-Pardoux (Haussmann and Pardoux, 1986) derives the identity stated below; for completeness we reproduce the argument at the end of the proof:

$$\int_s^t H_\tau \bullet \mathrm{d}\overline{W}_\tau = \int_s^t H_\tau \mathrm{d}\overline{W}_\tau + \int_s^t \mathrm{tr}(B_\tau) \mathrm{d}\tau. \tag{23}$$

Here, $\bullet \mathrm{d}\overline{W}_\tau$ denotes the backward integral. Consequently,

$$\int_s^t V(\tau) \nabla_x (r-G)(X_\tau^G, \tau)^\top \bullet \mathrm{d}\overline{W}_\tau = \int_s^t V(\tau) \nabla_x (r-G)(X_\tau^G, \tau)^\top \mathrm{d}\overline{W}_\tau + \int_s^t V^2(\tau) \Delta_x (r-G)(X_\tau^G, \tau) \mathrm{d}\tau.$$

Thus,

$$\int_s^t V(\tau) \nabla_x (r-G)(X_\tau^G, \tau)^\top \mathrm{d}W_\tau$$

$$= \int_s^t V(\tau) \nabla_x (r-G)(X_\tau^G, \tau)^\top \mathrm{d}\overline{W}_\tau - \int_s^t V^2(\tau) \nabla_x (r-G)(X_\tau^G, \tau)^\top \nabla \log Q_\tau(X_\tau^G) \mathrm{d}\tau$$

$$= \int_s^t V(\tau) \nabla_x (r-G)(X_\tau^G, \tau)^\top \bullet \mathrm{d}\overline{W}_\tau - \int_s^t V^2(\tau) \Delta_x (r-G)(X_\tau^G, \tau) \mathrm{d}\tau$$

$$- \int_s^t V^2(\tau) \nabla_x (r-G)(X_\tau^G, \tau)^\top \nabla \log Q_\tau(X_\tau^G) \mathrm{d}\tau. \tag{24}$$

Substituting (24) into (20) gives

$$\lim_{s \to t} \frac{1}{t-s} \left( \mathbb{E}_{\mathbb{P}^G} \left[ \exp \left( \int_s^t V(\tau) \nabla_x (r-G)(X_\tau^G, \tau)^\top \mathrm{d}W_\tau \right. \right. \right.$$

$$+ \int_s^t \left( \partial_\tau r(X_\tau^G, \tau) - \frac{1}{2} V^2(\tau) \|\nabla_x G(X_\tau^G, \tau)\|^2 + \nabla_x r(X_\tau^G, \tau)^\top (v(X_\tau^G, \tau) + V^2(\tau) \nabla_x G(X_\tau^G, \tau)) \right.$$

$$\left. + \frac{1}{2} V^2(\tau) \Delta_x r(X_\tau^G, \tau) \right) d\tau \bigg) \bigg| X_t^G = x \bigg] - 1 \bigg)$$

$$= \lim_{s \to t} \frac{1}{s - t} \left( \mathbb{E}_{\mathbb{P}^G} \left[ \exp \left( \int_t^T V(\tau) \nabla_x (r - G)(X_\tau^G, \tau)^\top \bullet d\overline{W}_\tau \right. \right. \right.$$

$$+ \int_s^t \left( \partial_\tau r(X_\tau^G, \tau) - \frac{1}{2} V^2(\tau) \|\nabla_x G(X_\tau^G, \tau)\|^2 + \nabla_x r(X_\tau^G, \tau)^\top (v(X_\tau^G, s) + V^2(\tau) \nabla_x G(X_\tau^G, \tau)) \right.$$

$$\left. + \frac{1}{2} V^2(\tau) \Delta_x r(X_\tau^G, \tau) + V^2(\tau) \Delta_x (G - r)(X_\tau^G, \tau) + V^2(\tau) \nabla_x (G - r)(X_\tau^G, \tau)^\top \nabla \log Q_\tau(X_\tau^G) \right) d\tau \bigg)$$

$$\bigg| X_t^G = x \bigg] - 1 \bigg)$$

$$=: \lim_{s \to t} \frac{1}{t - s} \left( \mathbb{E}_{\mathbb{P}^G} \left[ \exp \left( \int_t^T g(X_\tau^G, \tau) \bullet d\overline{W}_\tau + \int_t^T f(X_\tau^G, \tau) d\tau \right) \bigg| X_t^G = x \right] - 1 \right), \tag{25}$$

where $g(X_t^G, t) = V(t) \nabla_x (r - G)(X_t^G, t)^\top$ and

$$f(X_t^G, t) = \partial_t r(X_t^G, t) - \frac{1}{2} V^2(t) \|\nabla_x G(X_t, t)\|^2 + \nabla_x r(X_t^G, t)^\top \left( v(X_t^G, t) + V^2(t) \nabla_x G(X_t^G, t) \right)$$

$$+ \frac{1}{2} V^2(t) \Delta_x r(X_t^G, t) + V^2(t) \Delta_x (G - r)(X_t^G, t) + V^2(t) \nabla_x (G - r)(X_t^G, t)^\top \nabla \log Q_t(X_t^G).$$

By a Taylor expansion and the orthogonality between $dt$ and $d\overline{W}_t$,

$$\mathbb{E}_{\mathbb{P}^G} \left[ \exp \left( \int_s^t g(X_\tau^G, \tau) \bullet d\overline{W}_\tau + \int_s^t f(X_\tau^G, \tau) d\tau \right) \bigg| X_t^G = x \right] - 1$$

$$= \mathbb{E}_{\mathbb{P}^G} \left[ \int_s^t g(X_\tau^G, \tau) \bullet d\overline{W}_\tau + \int_s^t f(X_\tau^G, \tau) d\tau + \frac{1}{2} \left( \int_s^t g(X_\tau^G, \tau) \bullet d\overline{W}_\tau \right)^2 \bigg| X_t^G = x \right] + o(t - s)$$

$$= \mathbb{E}_{\mathbb{P}^G} \left[ f(X_t^G, t)(t - s) + \frac{1}{2} \|g(X_t^G, t)\|^2 (t - s) \bigg| X_t^G = x \right] + o(t - s),$$

where we used $\int_s^t g(X_\tau^G, \tau) \bullet d\overline{W}_\tau = 0$ and $(d\overline{W}_\tau)^2 = d\tau$. Substituting this expansion into (25) yields

$$\lim_{s \to t} \frac{\mathbb{E}_{\mathbb{P}^G} \left[ \exp \left( -\int_s^t V(\tau) \nabla_x G(X_\tau^G, \tau)^\top dW_\tau + \frac{1}{2} \int_s^t V^2(\tau) \|\nabla_x G(X_\tau^G, \tau)\|^2 d\tau + \int_s^t dr(X_\tau^G, \tau) \right) \bigg| X_t^G = x \right] - 1}{T - t}$$

$$= f(x, t) + \frac{1}{2} \|g(x, t)\|^2$$

$$= \partial_t r(x, t) - \frac{1}{2} V^2(t) \|\nabla_x G(x, t)\|_2^2 + \nabla_x r(x, t)^\top \left( v(x, t) + V^2(t) \nabla_x G(x, t) \right) + \frac{1}{2} V^2(t) \Delta_x r(x, t)$$

$$+ V^2(t) \Delta_x (G - r)(x, t) + V^2(t) \nabla_x (G - r)(x, t)^\top \nabla \log Q_t(x) + \frac{1}{2} \|V(t) \nabla_x (r - G)(x, t)\|_2^2.$$

Using $\nabla_x \log Q_t = \nabla_x \log(p_t e^{r(x,t)}) = \nabla_x \log p_t + \nabla_x r(\cdot, t)$, this simplifies to

$$\partial_t r - \frac{V^2(t)}{2} \left( \Delta r + \|\nabla r\|^2 \right) + \nabla r^\top \left( v + V^2(t) \nabla G \right) + V^2(t) \nabla \log p_t^\top \nabla (G - r) + V^2(t) \Delta G,$$

which establishes the desired identity. □

*Proof of Eq.* (23). By the definitions of the forward and backward Itô integrals, let $\Pi = (t_0, t_1, \ldots, t_n)$ be a partition of $[t, T]$ with mesh $|\Pi| = \max_{0 \le i \le n-1} |t_{i+1} - t_i|$. Then

$$\int_t^T H_s \bullet \mathrm{d}\overline{W}_s - \int_t^T H_s^\top \mathrm{d}\overline{W}_s = \lim_{|\Pi| \to 0} \sum_{i=0}^{n-1} (H_{t_{i+1}} - H_{t_i})(\overline{W}_{t_{i+1}} - \overline{W}_{t_i}) =: \langle H, \overline{W} \rangle_T - \langle H, \overline{W} \rangle_t,$$

where $\langle H, \overline{W} \rangle_s$ denotes their quadratic covariation at time $s$. Hence,

$$\int_t^T H_s \bullet \mathrm{d}\overline{W}_s - \int_t^T H_s^\top \mathrm{d}\overline{W}_s = \int_t^T \mathrm{d}\langle H, \overline{W} \rangle_s.$$

To evaluate right hand side, consider each coordinate, denoted by $H_s^i$ and $\overline{W}_s^i$. From the SDE for $H_s$, we have

$$\mathrm{d}H_s^i = (A_s)_i \mathrm{d}s + \sum_j (B_s)_{ij} \mathrm{d}\overline{W}_s^j,$$

where $(A_s)_i$ is the $i$-th component of $A_s$ and $(B_s)_{ij}$ the $(i,j)$-th entry of $B_s$. By Itô calculus,

$$\mathrm{d}\langle H^i, \overline{W}^i \rangle_s = \left\langle (A_s)_i ds + \sum_j (B_s)_{ij} \mathrm{d}\overline{W}_s^j, \mathrm{d}\overline{W}_s^i \right\rangle = \left\langle \sum_j (B_s)_{ij} \mathrm{d}\overline{W}_s^j, \mathrm{d}\overline{W}_s^i \right\rangle$$

$$= \sum_j (B_s)_{ij} \langle \mathrm{d}\overline{W}_s^j, \mathrm{d}\overline{W}_s^i \rangle = \sum_j (B_s)_{ij} \delta_{ij} \mathrm{d}s = (B_s)_{ii} \mathrm{d}s,$$

where $\delta_{ij}$ is the Kronecker delta. Summing over $i$ gives

$$\mathrm{d}\langle H, \overline{W} \rangle_s = \sum_i (B_s)_{ii} \mathrm{d}s = \mathrm{tr}(B_s) \mathrm{d}s.$$

Substituting back, we obtain

$$\int_t^T H_s \bullet \mathrm{d}\overline{W}_s = \int_t^T H_s^\top \mathrm{d}\overline{W}_s + \int_t^T \mathrm{tr}(B_s) \mathrm{d}s,$$

which proves Eq. (23). □

## C. Additional Implementation Details

**Gaussian Mixture Model.** For the primary GMM evaluation, we employ $N = 8,192$ particles and 500 discretization steps. Resampling is governed by an Effective Sample Size (ESS) criterion, triggered whenever ESS $< cN$ for a threshold parameter $c \in (0, 1)$. We set $c = 0.8$, following (Ren et al., 2025a).

In terms of the reward function, we impose a quadratic reward $r(x) = -\frac{1}{2}(x - \mu_r)^\top \Sigma_r^{-1}(x - \mu_r)$, which induces a posterior mixture with covariance $\widetilde{\Sigma} = (\Sigma_r^{-1} + (40I)^{-1})^{-1}$, component means $\widetilde{\mu}_i = \widetilde{\Sigma}((40I)^{-1}\mu_i + \Sigma_r^{-1}\mu_r)$, and weights $\widetilde{w}_i \propto \exp\left[-\frac{1}{2}(\mu_i - \mu_r)^\top (\Sigma_r + 40I)^{-1}(\mu_i - \mu_r)\right]$. This quadratic specification ensures that the posterior distribution remains analytically tractable, providing a rigorous ground truth for evaluation.

**Inverse Problems.** In inverse problems, we mainly test our methods and the baseline methods on the FFHQ-256 (Karras et al., 2019) dataset and ImageNet-256 (Deng et al., 2009) datasets. All images used for the tests in this paper are in RGB. For FFHQ-256, the 100 testing images are selected to be the first 100 images in the dataset, whose indexes range from 00000 to 00099. For ImageNet-256, the 100 testing images are selected to be the first 100 images in the ImageNet-1k validation set. Testing on this representative dataset can fully demonstrate the performance of the strategies.

**Text-to-Image Generation.** In text-to-image Generation, sampling follows classifier-free guidance (Ho and Salimans, 2022) with scale 7.5, alongside the DDIM sampler (Song et al., 2021) using $\eta = 1$ and $T = 100$ steps. The reward is defined as $r(X_t) = r(X_0 = \widehat{X}_t)$, where $\widehat{X}_t$ denotes the model's approximation of $\mathbb{E}_{p_{\text{data}}}[X_0 \mid X_t]$, and the guidance signal is set to $G(X_t) = \exp\left(10 \max_{0 \leq s \leq t} r(X_s)\right)$. The inference step we use is 100, and fixed resampling is employed with a resampling interval of 20, *i.e.* resampling is performed at $[0, 20, 40, 60, 80]$, following (Singhal et al., 2025).

As for Stable Diffusion model, we consider publicly available models fine-tuned for prompt alignment and aesthetic quality. Specifically, we consider DPO fine-tuned models for SD v1.5 and SDXL (Wallace et al., 2024; Rafailov et al., 2023). Approximately, SD v1.5 has 860M parameters and SDXL has 2.6B parameters. The substantial parameter scale of these backbones underscores the applicability of URGE to high-dimensional, large-scale generative tasks of practical significance.

## D. Additional Experimental Results and Discussions

In this section, we provide additional experimental results and more qualitative comparisons between URGE and existing baselines.

**Gaussian Mixture Model.** We first provide a visual comparison of the two-dimensional slices of all $N = 8,192$ particles in two specific dimensional for the main experiment. As shown in Fig. 4, PG produces visibly biased samples, while AFDPS still exhibits certain deviations from the analytical reference. In contrast, AFDPS+VCG, FK-Steering, and URGE accurately capture the correct modes and yield samples that align closely with the target distribution.

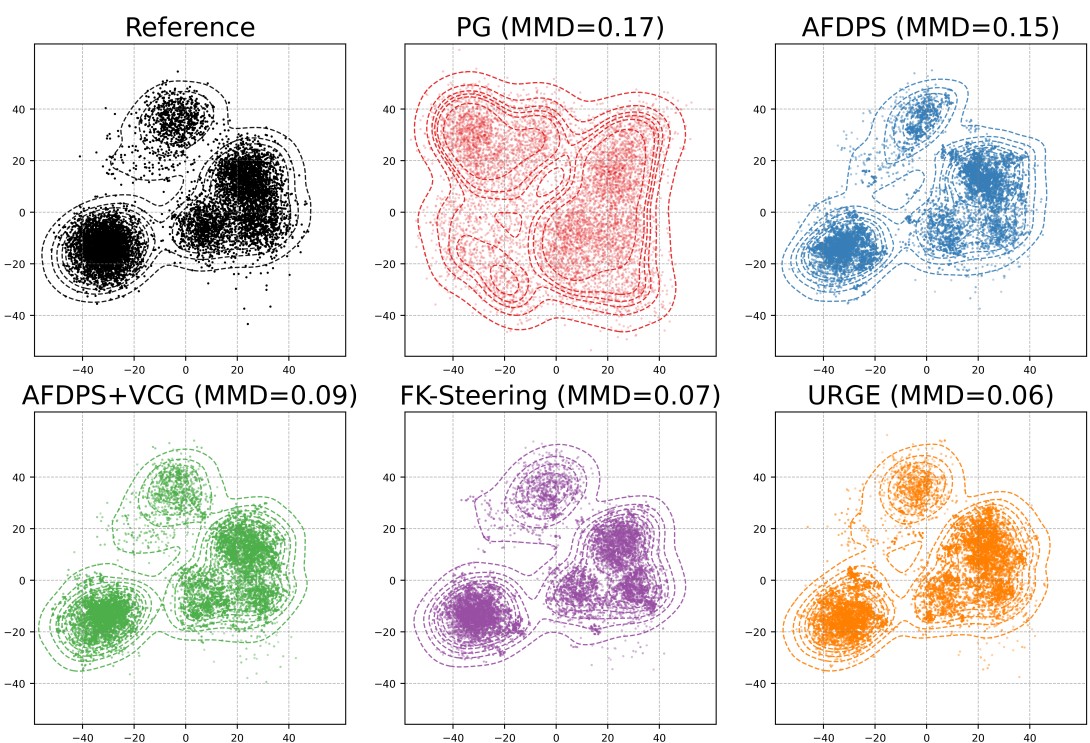

*Figure 4.* Qualitative comparison of inference-time scaling strategy on the Gaussian Mixture example.

In our main experiments, we use $K = 40$ components and $\text{Dim} = 30$ dimensions as the default setting. To further validate the robustness of URGE, we additionally conduct experiments with $K = 80$, $\text{Dim} = 30$ and $K = 40$, $\text{Dim} = 60$, respectively. We run experiments under three random seeds and report the averaged results, as table 6 and table 7 show. In the above setting, URGE continues to demonstrate stable and competitive performance, maintaining strong alignment with the analytical reference. It is worth noting that as the number of components and dimensionality increase, the gap between the performance of URGE and other strategies also increases, indicating that its advantage becomes more pronounced. This highlights that URGE has better scaling performance and may have stronger ability to handle increasingly complex problems.

| Method | MMD | SWD | Mean | Cov Frob |
|---|---|---|---|---|
| PG | 0.29 | 3.21 | 13.11 | 1029.85 |
| AFDPS | 0.12 | 1.55 | 6.94 | 679.62 |
| AFDPS+VCG | 0.10 | 1.46 | 6.78 | 658.96 |
| FK-Steering | 0.07 | 1.32 | 6.94 | 605.31 |
| **URGE** | **0.07** | **1.19** | **6.12** | **485.59** |

*Table 6.* Performance of different strategies on the Gaussian mixture toy example with $K = 80$, Dim $= 30$. Lower values indicate closer alignment with the analytical reference.

| Method | MMD | SWD | Mean | Cov Frob |
|---|---|---|---|---|
| PG | 0.324 | 3.127 | 20.815 | 1808.37 |
| AFDPS | 0.223 | 2.472 | 16.228 | 1369.89 |
| AFDPS+VCG | 0.133 | 1.873 | 13.716 | 1341.90 |
| FK-Steering | **0.092** | 1.561 | 12.843 | 1567.44 |
| **URGE** | 0.109 | **1.384** | **9.242** | **901.05** |

*Table 7.* Performance of different strategies on the Gaussian mixture toy example with $K = 40$, Dim $= 60$. Lower values indicate closer alignment with the analytical reference.

Finally, we investigate the sensitivity of the methods to the number of discretization steps. While our main experiments use 500 steps, we fix the random seed and evaluate all methods at steps of $[125, 250, 500, 1000, 2000]$. The results are plotted in Figure 5. Across these discretizations, URGE remains the most consistent performer, achieving the best in the majority of cases and exhibiting modest variation. In contrast, competing strategies show unpredictable fluctuations across step sizes. This empirical stability indicates that URGE is robust to the discretization and generalizes more reliably.

**Inverse Problems.** To further assess the generality of URGE, we conducted experiments on FFHQ-256, aiming to verify its effectiveness on more models. Table 8 summarizes PSNR and LPIPS for representative tasks. URGE consistently tracks the behavior of AFDPS and achieves results of comparable quality. This alignment is significant, as it demonstrates that URGE preserves the high performance of derivative-based methods while maintaining robust stability.

We also measure inference time on the Gaussian Deblurring task. In Figure 6, AFDPS-ODE exhibits substantially larger cost across all numbers of particles, whereas URGE exhibits the most favorable computational scaling and consistently achieves the shortest inference times. These timing results support that URGE attains a more practical balance between particle budget and runtime, making it particularly attractive when computational efficiency is a concern.

For a more intuitive visual comparison, we select several portrait examples from the inverse problems experiments and present representative reconstructions on FFHQ in Figures 10 and 11. The visual results confirm that AFDPS-SDE, AFDPS-ODE and URGE all produce strong restorations. Even more, in several cases URGE yields visibly improved reconstructions compared to AFDPS, further supporting the effectiveness of URGE.

Ultimately, we focus on the runtime of URGE. Following (Chen et al., 2025), we measure the runtime with AFDPS-ODE run at $N = 5$ particles to offset the additional computational cost introduced by the corrector step, while all other methods are evaluated at $N = 10$ particles. As shown in Figure 8, across different inverse tasks and models, URGE exhibits excellent runtime performance and is generally faster than the baselines, demonstrating that its practical computational cost is competitive in addition to its reconstruction quality.

**Text-to-Image Generation.** The scaling of the diffusion model as the number of particles increases is a notable issue. We measure ImageReward under the same experimental settings as the main experiment while varying the number of particles. Figure 7 shows the ImageReward curves for URGE and FK-Steering, both evaluated with SDv1.5. As particle counts increase, URGE's ImageReward grows faster and consistently outperforms FK-Steering across all tested numbers of particles.

We further present a visual comparison on prompts that describe two colored objects, since color requires correct interpretation of object ordering. As shown in Table. 9, all other experimental settings follow the main experiment except

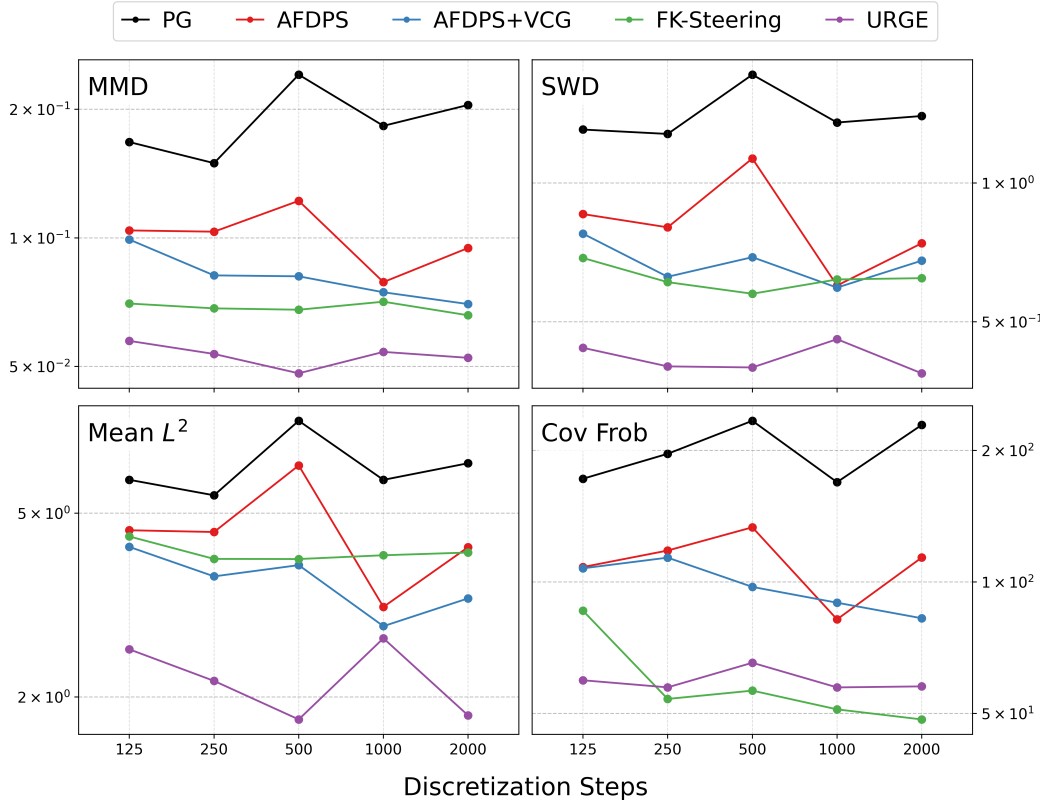

*Figure 5.* Performance metrics versus discretization steps on Gaussian Mixture Model.

for the prompts used. The selected prompts contain exactly two colored entities. On colored two-object prompts, URGE also performs strongly. The generated samples adhere more strictly to the specified color attributes and spatial ordering, demonstrating enhanced compositional fidelity.

Similar to additional experiments on other tasks, we examine the scaling of inference time with the number of particles. Figure 9 shows the comparison of the inference time between URGE and FK-Steering using gradient guidance. URGE not only has shorter inference time at all particle counts, but also has slower scaling with increasing number of particles, showing its practical inference time performance. In summary, URGE outperforms FK-Steering in both inference-time efficiency and quantitative alignment metrics.

| Method | Gaussian Deblurring | | Motion Deblurring | | Super Resolution | | Box Inpainting | |
|---|---|---|---|---|---|---|---|---|
| | PSNR↑ | LPIPS↓ | PSNR↑ | LPIPS↓ | PSNR↑ | LPIPS↓ | PSNR↑ | LPIPS↓ |
| SGS-EDM | 24.37 | 0.2833 | 22.18 | 0.3593 | 15.81 | 0.4208 | 22.18 | 0.2698 |
| FK-Corrector | 21.22 | 0.4039 | 20.49 | 0.4284 | 20.64 | 0.4133 | 16.96 | 0.5503 |
| AFDPS-SDE | 24.76 | 0.2590 | 23.56 | 0.2866 | **22.99** | **0.3033** | 25.37 | 0.2089 |
| AFDPS-ODE | **24.96** | **0.2571** | 23.56 | 0.2901 | 21.45 | 0.3347 | **25.59** | **0.1968** |
| **URGE** | 24.79 | 0.2593 | **23.58** | **0.2865** | 22.96 | 0.3056 | 25.31 | 0.2092 |

*Table 8.* Results on 4 inverse problems for 100 validation images from FFHQ-256.

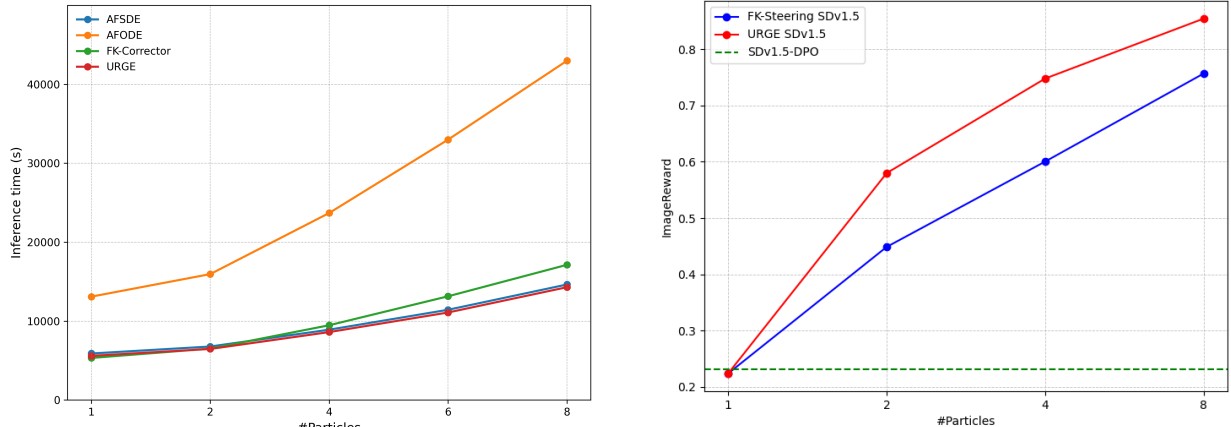

*Figure 6.* Inference time versus number of particles for the Gaussian Deblurring task.

*Figure 7.* ImageReward versus number of particles for URGE and FK-Steering (SDv1.5).

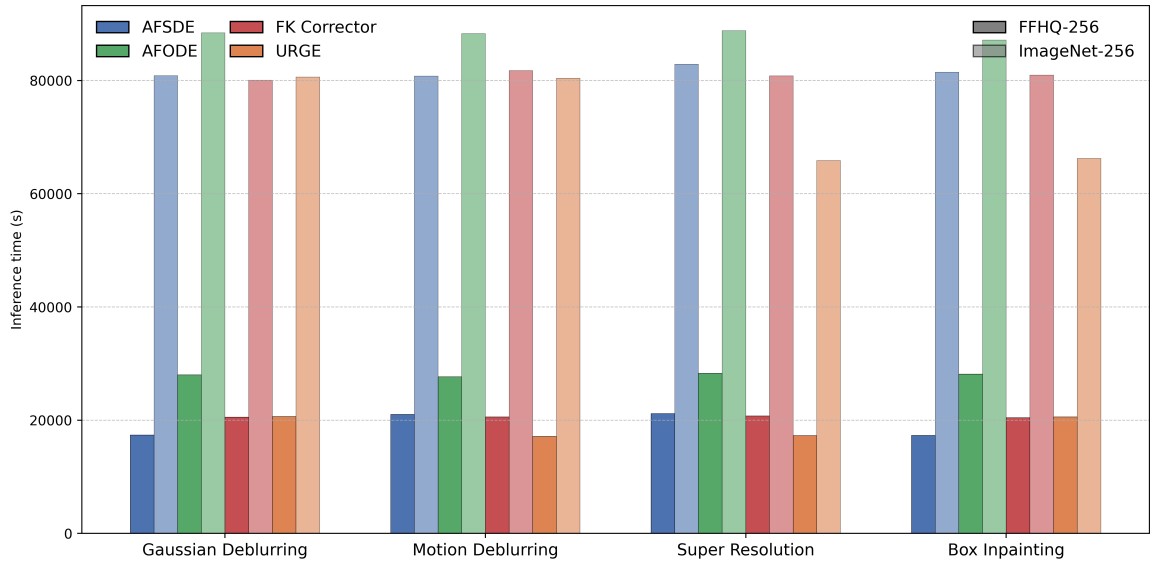

*Figure 8.* Inference time for the inverse problems tasks on FFHQ-256 and ImageNet-256. Results report runtimes where AFDPS-ODE is run with $k = 5$ particles while all other methods use $k = 10$ particles.

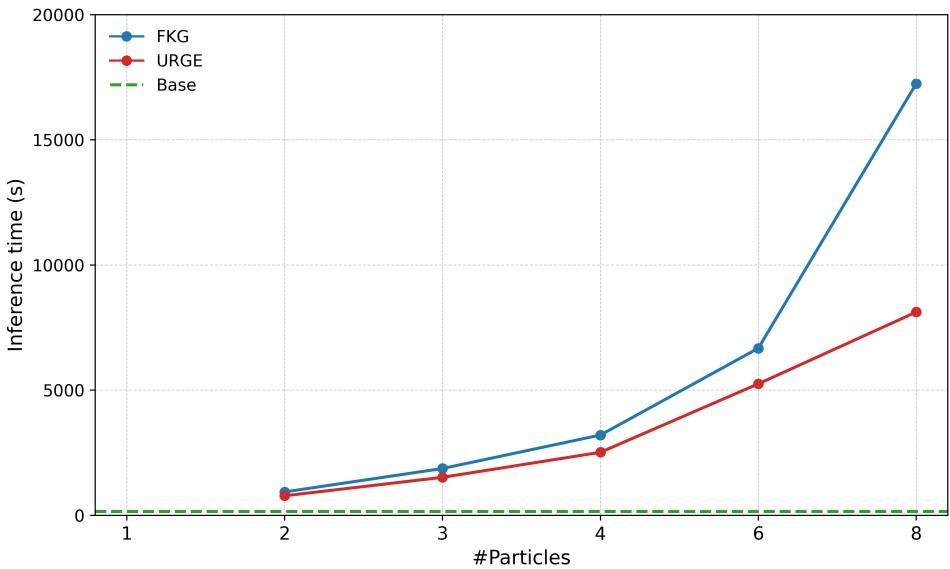

*Figure 9.* Inference time for the text-to-image tasks on 20 prompts. `FKG` refers to `FK-Steering` with gradient guidance. Base is ran with only 1 particle.

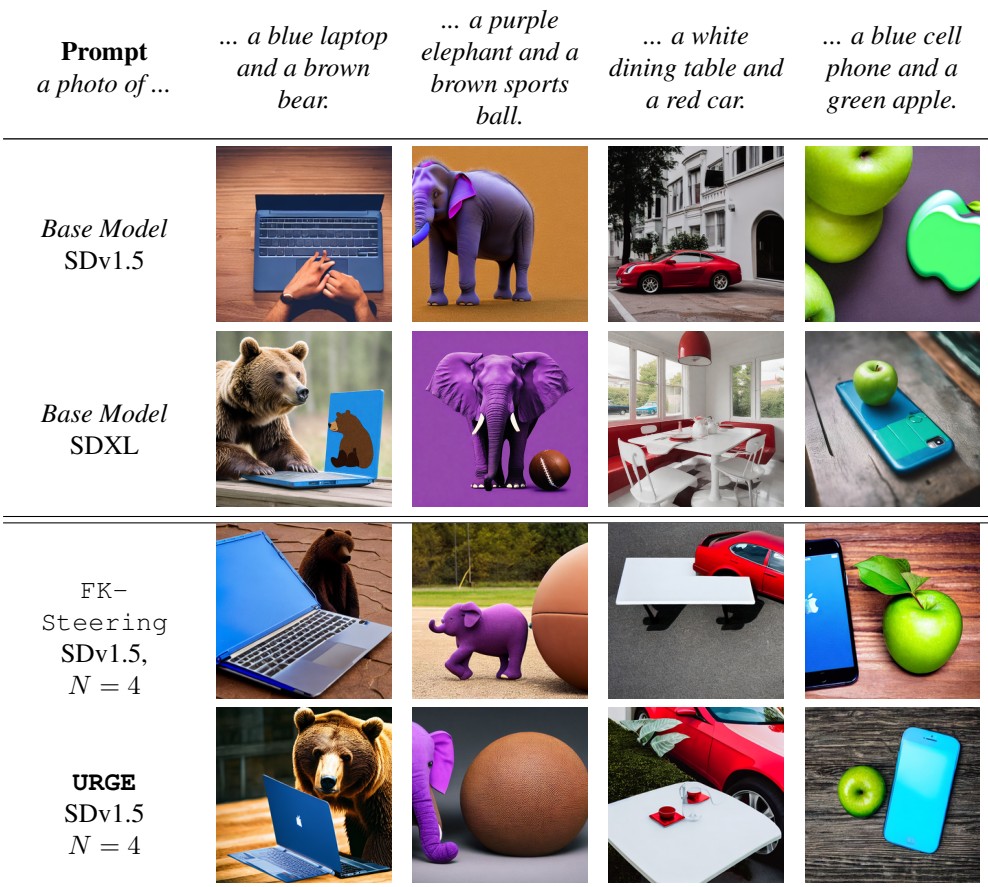

*Table 9.* Comparison of three methods under four colored prompts. URGE, even using SDv1.5, often matches or approaches the effect of the baseline that uses SDXL, and clearly outperforms FK, with results that better match the text and common sense.

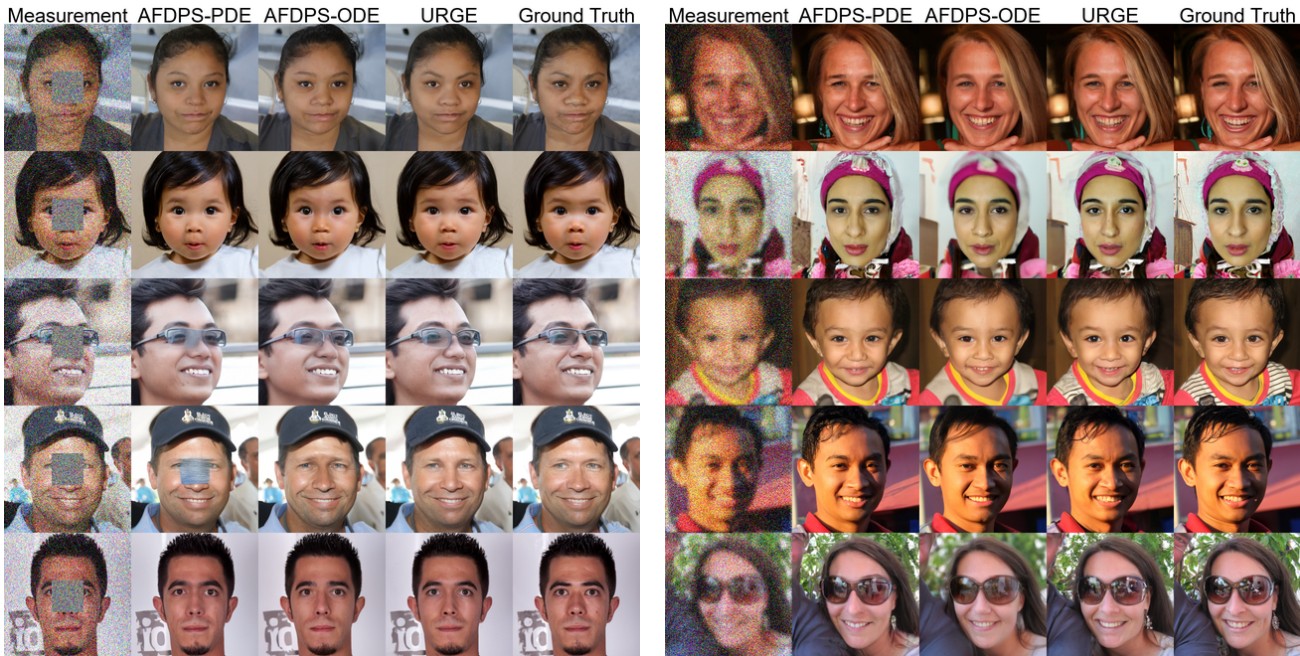

*Figure 10.* Additional visual examples for the Box inpainting problem and the Gaussian deblurring problem on FFHQ.

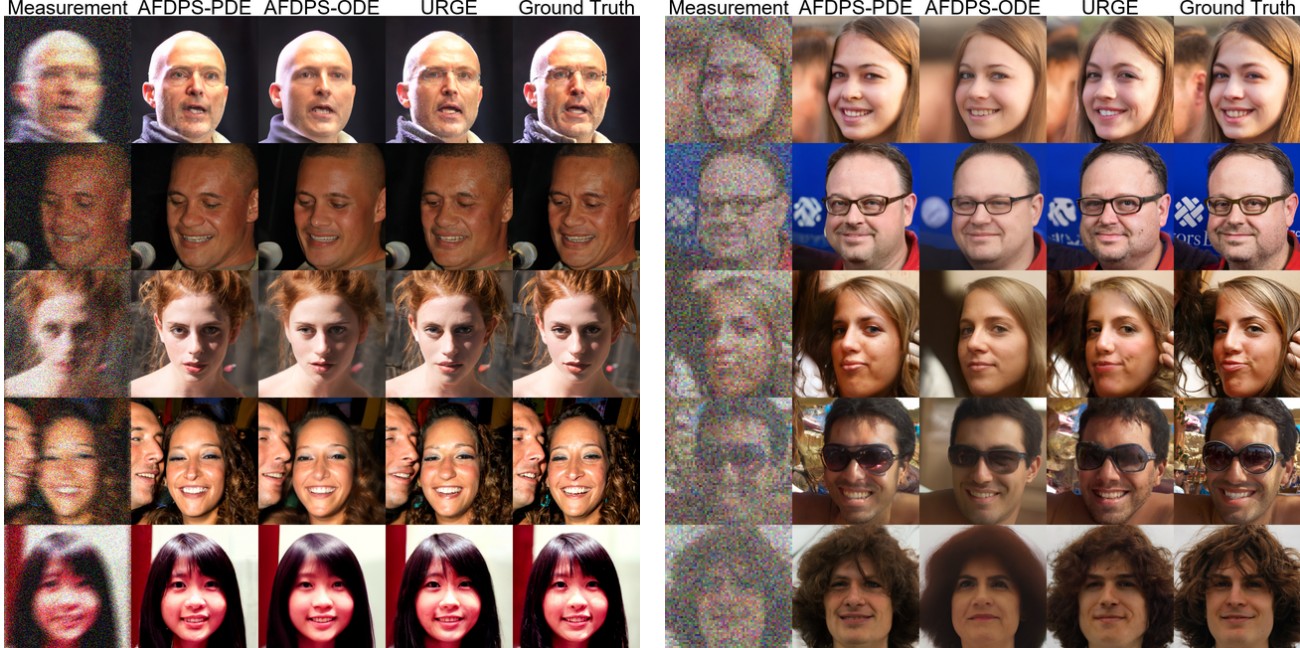

*Figure 11.* Additional visual examples for the Motion deblurring problem and the Super resolution problem on FFHQ.

