# OpenReview forum: "Simple Approximation and Derivative Free Inference-Time Scaling for Diffusion Models via Sequential Monte Carlo on Path Measures"
_ICML.cc/2026/Conference — ICML 2026 regular_

### Official Review · Reviewer_fLu9 · 2026-03-12

**Soundness:** 3
**Presentation:** 2
**Significance:** 3
**Originality:** 3
**Overall Recommendation:** 4
**Confidence:** 4

**Summary:**

This paper aims to provide a derivative-free inference-time algorithm for alignment. The authors adopt the Girsanov theorem to calculate the pathwise reweighting term. They prove an equivalence between pathwise and particle-wise SMC. They verify the effectiveness of the proposed method on simulations, inverse problem, and text-to-image generation tasks.

**Compliance With Llm Reviewing Policy:**

Affirmed.

**Final Justification:**

The strengths of this paper mainly lie in originality and clarity. During the rebuttal, the authors provide some quantitative results on one benchmark, which somehow improves the significance.

**Key Questions For Authors:**

Could the authors provide quantitative results on GenEval?

**Strengths And Weaknesses:**

Strengths:

1. Utilizing the measure to calculate the importance reweighting is novel.
2. The theoretical equivalence between path-space importance weight and particle-space reweighting term is novel.

Weaknesses:
1. Although the paper claims that they are derivative-free, they still use $\nabla_x G(X^G_t, t)$. In the experiments, the authors adopt $G=r$.
2. In the literature of training-free guidance, the reward function is always used to calculate the quality of the final output $x_0$. To calculate the reward of $x_t$, they would use $\mathbb{E}_{p(x_0|x_t)}[r(x_0)]$. However, the authors adopt a time-dependent reward function without training the network. $r(x, t) = g(t)r(x)$ is a questionable design and may lead to a biased solution to the origin alignment problem.
3. (Minor) Although (Chidambaram et al., 2024) proves for some of the reference processes, guidance does not lead to a reward-tilted path measure, the guidance is unbiased for some reference processes. It cannot be used as evidence to prove that incorporating this drift modification is a generally bad method.

---

> ### Author Rebuttal · Authors · 2026-03-31
>
> Thank you for the thoughtful review. We are pleased that the reviewer found our method novel and  theoretical equivalence between path-space importance weight and particle-space reweighting term interesting. The lifting to path space significantly simplifies practical implementation and improves inference speed. We address the reviewer's comments point by point.
>
> ### regards derivative-free
>
> The derivative of the reward is only used in construction of the guidance but not in the reweighting procedure we designed. (i.e. URGE is derivative free in the correction/reweighting step) In general guidance may not need to be a gradient, In our new revision (e.g. [1]), we use G function to represent the guidance and then our method is totally derivative free. Sorry for the potential confusion. FK-corrector/AFDPS also need the second order derivative of the reward  and the score function  while our method doesn’t.
>
> ### regards the reward function
>
> Yes the reward function should be computed as expected future reward. However, once you use the reward function as , it’s hard to compute the gradient of the function, thus in the literature people use the original reward function as a proxy to design the guidance procedure. One possible alternative is using Doob’s h-Transform  [1], which is published after our submission. We leave this as future work to consider more advanced guidance techniques. At the same time, we treat the guidance function $G$ purely as a general proposal mechanism, not as part of the target distribution itself. The target distribution remains the terminal reward-tilted law, and the role of the Radon–Nikodym correction in URGE is precisely to account for the mismatch between the proposal and the target. Consequently, URGE can be used to correct arbitrary guidance functions.
>
> ### regards limitation and potential negative social impact
>
> We thank the reviewer for pointing this out. We agree that the current draft does not sufficiently discuss limitations and broader societal impact, and we will revise the paper to make these aspects explicit. Our method is an inference-time scaling approach that reweights and resamples multiple trajectories to better match a reward-tilted target distribution, rather than introducing a new generative model class. Nevertheless, it does have important limitations: it increases inference-time compute due to multi-particle simulation and repeated resampling, and remains dependent on the quality of the user-specified reward/objective. In particular, if the reward is misspecified or encodes undesirable preferences, the method may amplify those biases by more effectively steering sampling toward that target. On broader impact, while we view URGE as a general-purpose inference-time correction method, stronger inference-time control could also be misused to generate more persuasive or better-optimized synthetic content. We will add a dedicated discussion of these limitations and societal considerations in the revision.
>
> ### Regards Best-of-N baseline
>
> We added a Best-of-N baseline, ∇(N = 4), to the paper and show that URGE significantly outperforms this naive Best-of-N approach.
>
> | Sampler | Clip-Score ↑ | HPS ↑ | IR ↑ |
> |---|---:|---:|---:|
> | N = 1 | 0.2730 | 0.2619 | 0.2144 |
> | ∇(N = 1) | 0.2725 | 0.2614 | 0.2125 |
> | FK (N = 4) | 0.2901 | 0.2849 | 0.8397 |
> | ∇(FK, N = 4) | 0.2899 | 0.2839 | 0.7896 |
> | **URGE (N = 4)** | **0.2997** | **0.2927** | **0.9955** |
>
> Thanks again for your efforts in reviewing this paper. We hope we have addressed your concerns regarding the role of reward derivatives, the use of a time-dependent proxy reward for practical guidance, and the limitations/broader impact discussion. We are more than happy if you can reconsider the review rating.
>
> [1] Zhu, Qijie, et al. “Training-Free Adaptation of Diffusion Models via Doob's $h$-Transform”. arXiv preprint arXiv:2602.16198, 2026.

---

> > ### Author Rebuttal · Reviewer_fLu9 · 2026-04-04
> >
> > Thank you for the response. However, I must point out that the authors have not addressed my actual concerns. Specifically, the response regarding "limitation and potential negative social impact" and "Best-of-N baseline" were not part of my questions. I would kindly request the authors to address my question on the time-dependent reward function and quantitative results on GenEval.

---

> > > ### Author Response · Authors · 2026-04-05
> > >
> > > We thank the reviewer for the follow-up, and we are sorry our last message did not answer your questions directly. Below, we address the time-dependent reward and GenEval, as you asked.
> > >
> > > ## Regards the time-dependent reward
> > >
> > > The time-dependent reward does not introduce bias. In our algorithm, the distribution of \(X_t\) is tilted by \(r(x,t)\), whereas the final target is the distribution of \(X_1\), which is exactly tilted by \(r(x)\). This is guaranteed when we choose the schedule \(r(x,t)=g(t)r(x)\) with \(g(1)=1\). The purpose of introducing the time-dependent reward is to gradually guide the dynamics, making the resampling weights closer to uniform (since we gradually change the distribution) and thus reducing the variance of the resampling step. We will add a remark after Theorem 3.1 to clarify this point.
> > >
> > > ## Regards quantitative results on GenEval
> > >
> > > It takes us some time to run the experiment on GenEval, and here is the result. We add GenEval next to Clip-Score, HPS, and ImageReward. N=1 and gradient-guided N=1 use best-of-10 seeds per metric. FK, gradient-guided FK, and URGE use N=4 particles; GenEval averages the per-run max over 3 seeds. Here are the results.
> > >
> > > | Sampler | Clip-Score ↑ | HPS ↑ | IR ↑ | GenEval ↑ |
> > > |---------|-------------:|------:|-----:|----------:|
> > > | N = 1 | 0.2730 | 0.2619 | 0.2144 | 0.6400 |
> > > | ∇(N = 1) | 0.2726 | 0.2617 | 0.2074 | 0.6400 |
> > > | FK (N = 4) | 0.2901 | 0.2849 | 0.8397 | 0.7200 |
> > > | ∇(FK, N = 4) | 0.2899 | 0.2839 | 0.7906 | 0.7467 |
> > > | URGE (N = 4) | 0.2997 | 0.2927 | 0.9955 | 0.7800 |
> > >
> > > On GenEval, URGE still achieves the highest score in this table, so the compositional benchmark aligns with the other metrics.
> > >
> > > We apologize again for the earlier reply drifting from your questions. Thank you for the careful read. We are more than happy if you can reconsider the review rating.

---

### Official Review · Reviewer_sLPz · 2026-03-13

**Soundness:** 3
**Presentation:** 3
**Significance:** 3
**Originality:** 3
**Overall Recommendation:** 4
**Confidence:** 3

**Summary:**

This paper focuses on inference-time scaling for diffusion models and proposes a derivative-free guidance method based on Sequential Monte Carlo (SMC) on path measures. The authors introduce a method called Unbiased Resampling via Girsanov Estimation (URGE), which performs pathwise importance reweighting using the Girsanov change of measure to enable derivative-free inference-time scaling. The paper provides theoretical analysis showing the correctness of the proposed pathwise weighting scheme and its connection to particle-level weighting. Empirically, the effectiveness of the method is demonstrated through experiments on toy problems, inverse problems, and text-to-image generation tasks.

**Compliance With Llm Reviewing Policy:**

Affirmed.

**Final Justification:**

The paper proposes a novel, theoretically grounded method based on SMC and Grisanov weighting. The formulation is clear and the theoretical analysis is solid. The authors' response has addressed most of my concerns. While some empirical evaluations could be strengthened, the overall contribution remains meaningful, and I maintain my positive weighting.

**Key Questions For Authors:**

Please provide responses to the concerns raised in the Weaknesses section.

**Limitations:**

The paper does not explicitly discuss limitations and potential negative societal impact.

**Strengths And Weaknesses:**

### [Strengths]

1. Inference-time scaling for diffusion models has recently attracted significant attention, and proposing improved methods in this direction is both timely and meaningful.

2. The proposed method is well motivated and theoretically grounded. The formulation of the method is clearly presented, and the theoretical analysis provides useful insights into the properties of the algorithm.

3. The overall framework is clearly formulated, and the role of each component in the algorithm is well explained. The theoretical results further help characterize the behavior and guarantees of the proposed method, which strengthens the credibility of the approach.

### [Weaknesses]

1. Since the proposed method relies on multiple particles, understanding the trade-off between performance and computational cost is important. Reporting runtime and GPU memory usage alongside performance metrics would help clarify the practical efficiency of the method relative to existing approaches.

2. In the inverse problem experiments, the performance differences between methods appear relatively small. Evaluating the methods under comparable computational budgets (e.g., equal runtime or memory constraints) could provide a clearer picture of the practical benefits of the proposed approach. Also, several experimental results do not report error bars or statistical variability. Including error bars or multiple runs would improve the reliability and interpretability of the reported results.

3. Although the paper discusses methods such as FK-Corrector and AFDPS, they are not compared in the text-to-image generation experiments due to claimed inapplicability. However, providing comparisons, including runtime and GPU memory consumption, could further highlight the practical advantages of the proposed approach.

4. For the text-to-image generation experiments, it would be informative to compare against a Best-of-N sampling baseline. Such a comparison would help establish a computational baseline corresponding to generating multiple samples and selecting the best one, making it easier to quantify the additional gain provided by the proposed method.

5. Most experiments appear to be conducted on SDv1.5. Evaluating the method on more recent and larger diffusion models (e.g., SD3, FLUX) would better demonstrate the practical applicability and scalability of the approach.

---

> ### Author Rebuttal · Authors · 2026-03-31
>
> We thank the reviewer for the constructive feedback and for acknowledging the timeliness and solid theoretical grounding of our work. We agree that better clarifying the practical trade-offs, reporting statistical variability, and strengthening baselines will substantially improve the paper.
>
> ### Regards the trade-off between performance and computational cost
>
> Numerically, we benchmark the trade-off between performance and the number of discretization steps or particles on synthetic examples in Figures 2 and 5, and for prompt-to-image generation in Figures 3 and 9. We apologize for not presenting this more clearly. This also leads to an interesting theoretical question. When the step size is smaller, the resampling weights become closer to 1, which results in a smaller resampling error. More quantitatively, if the step size is \(O(\Delta t)\), then the resampling variance is also \(O(\Delta t)\), which helps control the total variance at the final time. This leads to the optimal scaling law that the number of particles should be proportional to the number of steps. We agree that better guidance may further reduce the variance. This is, in fact, another motivation for our analysis of AFDPS.  Our analysis suggests that the main difficulty in AFDPS is controlling the variance introduced by the resampling step. In the special case where the function \(G\) is a solution to a Bellman equation (which can be simulated using Monte Carlo methods [1]), It\^o's lemma implies that the reweighting factor is exactly 1. More generally, the magnitude of the log reweighting factor can be bounded in terms of the Bellman error. This analysis also yields a new scaling law that balances the Monte Carlo cost of simulating Doob’s h-transform, the number of particles, and the number of generation steps.
>
>
> ### Regards missing FK-Corrector and AFDPS baseline for text-to-image generation
>
> FK-Corrector and AFDPS require higher-order gradients of the reward. In the text-to-image generation setting, the reward is given by a neural network, for which computing higher-order gradients is difficult in high-dimensional spaces. For this reason, the FK-Corrector and AFDPS papers also do not implement their methods for text-to-image generation. We view this as a flexibility advantage of our method. We will clarify this point in the revision and apologize for any confusion.
>
> ### Regards the performance gain in Inverse Problem
> Yes, the performance gain on inverse problems appears marginal in terms of PSNR and LPIPS. **However, AFDPS requires higher-order gradients of the reward, whereas URGE does not. In other words, URGE uses less information and compute while still achieving comparable performance.** Moreover, for tasks such as super-resolution and box inpainting, URGE is substantially faster, as shown in Figure 8. We will clarify this point in the revision and apologize for any confusion.
>
>
> ### Regards best-of-N baseline
>
> We implement a best-of-N baseline following the reviewer’s suggestion. Concretely, we run 10 random seeds for the base sampler and its gradient-guided variant, and for each metric we report the best value achieved across seeds. The current resulting table is shown below.
> | Sampler | Clip-Score ↑ | HPS ↑ | IR ↑ |
> |---|---:|---:|---:|
> | N = 1 | 0.2730 | 0.2619 | 0.2144 |
> | ∇(N = 1) | 0.2725 | 0.2614 | 0.2125 |
> | FK (N = 4) | 0.2901 | 0.2849 | 0.8397 |
> | ∇(FK, N = 4) | 0.2899 | 0.2839 | 0.7896 |
> | **URGE (N = 4)** | **0.2997** | **0.2927** | **0.9955** |
> Even with this best-of-N selection for the base sampler and its gradient-guided variant, \texttt{URGE} remains the top-performing method on the reported metrics in this comparison. Best-of-N is a necessary baseline because it provides a fair reference point for evaluating methods. We also reran most of the experiments multiple times and updated the draft accordingly. After adding error bars, the results still show clear improvements from our method.
>
> Thanks again for your efforts into reviewing this paper. We hope we have addressed your concerns regarding the compute--performance trade-off, the lack of Best-of-$N$ and other baselines, the absence of error bars/statistical variability, and the limitations/broader impact discussion. We are more than happy if you can reconsider the review rating.

---

> > ### Author Rebuttal · Reviewer_sLPz · 2026-04-01
> >
> > Thank you for the authors’ response. Most of my concerns have been well addressed, and I would like to maintain my positive rating at now. One point I would like to clarify is regarding my comment on the limited performance gap in the inverse problem setting. As you mentioned, I understand that there is a significant difference in computational cost, and my comment was intended from the perspective that this could be demonstrated more effectively. In particular, comparing performance under the same computational budget (e.g., runtime or memory) might make the performance gap more evident. Separately, from my understanding, the FK-Corrector paper seems to include image generation experiments using EDM2 and SDXL (e.g., Section 5.1 and Section F.5). Therefore, I am curious about the reason behind the statement that "FK-Corrector and AFDPS papers also do not implement their methods for text-to-image generation."

---

> > > ### Author Response · Authors · 2026-04-03
> > >
> > > We thank the reviewer for catching this.
> > >
> > > **Regards Inferecne Time for Inverse Problem**  For tasks such as super-resolution and box inpainting, where gradient computation is particularly time-consuming, URGE is substantially faster, as shown in Figure 8. We apologize for not making this point clear in the previous version. To address this, we will move Figure 8 to the main text and expand the discussion accordingly.
> > >
> > > **Regarding the text-to-image generation experiments**  Appendix F.5 of the FKC paper reports no clear improvement of SDXL+FKC over vanilla CFG, and the gains are not consistent across randomly sampled prompts. This suggests that FKC does not reliably outperform standard CFG in this setting. In addition, the authors’ public code still marks the text-to-image CFG component in Section 5.1 as “coming soon,” so a complete public implementation is not yet available. At the same time, the main text does not clearly specify the reward function used in the experiments, nor does it provide sufficient experimental details, such as how the Laplace approximation is computed for a large network. As a result, reproducing these experiments is difficult for us.
> > >
> > > We hope this answers your question. Thank you again for the careful read.

---

### Official Review · Reviewer_FaxC · 2026-03-13

**Soundness:** 4
**Presentation:** 3
**Significance:** 4
**Originality:** 3
**Overall Recommendation:** 5
**Confidence:** 3

**Summary:**

This work focus on inference-time guidance for diffusion models. The major contribution of this work is to use  the Girsanov change of measure to apply pathwise importance weights to diffusion trajectories, leading to a derivative-free SMC weights. The authors then  establish rigorous theoretical guarantee on the equivalence between path-space resampling (URGE) and particle-space Sequential Monte Carlo (SMC) methods like FK-Corrector and AFDPS.

**Compliance With Llm Reviewing Policy:**

Affirmed.

**Key Questions For Authors:**

- The quality of SMC can be measured through the variance of the weights, how does the variance of URGE's path weights compare to AFDPS's particle weights? Is there any theoretical investigation on this?

**Limitations:**

I'd suggest the authors, in the revision,  to explict discuss the points I mentioned in the weakness.

**Strengths And Weaknesses:**

Strength:

- Theoretical Soundness and Unification: I have no reservations regarding the mathematical soundness of this work. More importantly, the application of the Girsanov theorem to derive trajectory weights is mathematically clean and elegantly presented. Lemma 3.2 and Thm 3.3 represent strong theoretical contributions; they successfully establish a marginalized equivalence between path-space resampling and particle-space reweighting, providing a unified theoretical understanding that bridges URGE with prior methods like AFDPS.

- Although resampling trajectories based on Radon-Nikodym ratios of path measures is standard in classical SMC literature, its application to diffusion guidance is highly novel and elegant in this context. By lifting the weights to the path space, the algorithm naturally bypasses the need to compute spatial derivatives (hence avoid evaluating the Laplacian of the reward and the score of the marginal distribution of diffusion model). This significantly simplifies practical implementation and boost inference speed.

- The presentation of this work is clear and easy to follow.

Weakness:

- Missing analysis on discretization bias:  While Theorem 3.1 proves unbiasedness in the continuous limit , the practical algorithm would require numerical discretization. Unless I'm missing something, this work does not quantify the introduced bias from discretization. Consequently, the practical, discrete-time implementation is not truly asymptotically exact. It would be ideal to have an error bound on the discretization bias or explicitly contrast their guarantees with literature that targets exact discrete-time conditional sampling (e.g., Wu et al., 2023, Practical and asymptotically exact conditional sampling in diffusion models).

- The efficiency of the SMC resampling relies heavily on the quality of the proposal measure---in this case, determined by the choice of the guidance $G$. This work lacks dedicated discussion or theoretical investigation on how URGE would perform as the quality of G varies.

- Although derivative-free, URGE still requires simulating and storing $N$ parallel trajectories by the nature of SMC. Due memory constraints, simulating a large number of particles simultaneously for models like SDXL is highly restrictive. While classical SMC theory relies on the asymptotic regime of large $N$ for its guarantees, this is practically infeasible here. Although this is a generic problem for diffusion SMC rather than a flaw specific to URGE, I'd recommend discussing the computational bottleneck more transparently.

Minor issues:
- Missing discussion on inference-time control based on parallel tempering (PT):  CREPE: Controlling Diffusion with Replica Exchange (He et al., 2025). While URGE tries to fix SMC's limitations by altering how the weights are calculated,  CREPE argues the favor of PT to solve the inherent memory bottlenecks and particle degeneracy issues of SMC. Also, the Girsanov change of measure for getting weights of particles is precisely used in CREPE (termed Radon-Nikodym estimators).


- Sloppy bibliograph section, to list a few:
  - Misspelled First Author (Line ~620) the first author of "Practical and asymptotically exact conditional sampling in diffusion models" is Luhuan Wu instead of Hao Wu.
  - Several published work cited as Arxiv preprints, e.g., Song et al. (2020), Lipman et al. (2022), Podell et al. (2023), the Feyman-Kac corrector, etc.

---

> ### Author Rebuttal · Authors · 2026-03-31
>
> Thank you for the thoughtful review. We are pleased that the reviewer found our application to diffusion guidance to be highly novel and elegant in this context, and that the benefits of lifting to path space were clear. This formulation significantly simplifies practical implementation and improves inference speed, and we are glad the presentation was easy to follow. **Regarding the computational bottleneck for diffusion SMC, we have added a discussion on its practical application in our new version, demonstrating that its efficiency shall be affected by the number of particles when we discuss the inference time scaling figures (Figure 2 and 3).** Regarding the minor points, we have corrected the typo noted in the original submission and apologize for any confusion it may have caused. In addition to the citation issue, we also identified several typos in the main text that were introduced under time constraints. We have since prepared a substantially improved revised version with these issues carefully corrected.**We have also included parallel tempering methods such as CREPE. We apologize for the earlier omission of this discussion.**
> ### Regards Discritization Error and Choice of Guidance
> Quantitatively, the bias of discretization is of order $O(\Delta t)$, which follows from the standard weak error theory for time discretizations of SDEs with the Euler--Maruyama scheme [1,2]. As the step size decreases, the resampling weights remain closer to one, leading to reduced resampling error. More precisely, when the step size is $O(\Delta t)$, the resampling variance is also of order $O(\Delta t)$, which helps control variance growth over time. Consequently, the component of the variance associated with time discretization remains $O(1)$ and does not accumulate as $\Delta t \to 0$. On the other hand, the dependence on the number of particles $N$ follows the standard Monte Carlo scaling: the variance is $O(N^{-1})$, equivalently the root-mean-square error is $O(N^{-1/2})$ [3]. Therefore, the total mean-square error can be decomposed as $\mathrm{MSE}^2[\texttt{URGE}] = O(\Delta t) + O(N^{-1})$. This would lead to  selecting the number of particles proportion to the time steps.
> We agree that better guidance may further reduce the variance. This is, in fact, another motivation for our analysis of AFDPS.  Our analysis suggests that the main difficulty in AFDPS is controlling the variance introduced by the resampling step. In the special case where the function $G$ is a solution to a Bellman equation (which can be simulated using Monte Carlo methods [4]), Ito's lemma implies that the reweighting factor is exactly 1. More generally, the magnitude of the log reweighting factor can be bounded in terms of the Bellman error. This analysis also yields a new scaling law that balances the Monte Carlo cost of simulating Doob’s $h$-transform, the number of particles, and the number of generation steps.
> Formalizing these observations rigorously would require a substantial amount of additional analysis, likely beyond the scope of the current paper. We therefore leave this direction for future work and will add a discussion of it in the revised version.
>
>
> ### Regards Comparison of Variance of Weights for Different Methods
> The variance of the AFDPS weight is smaller than that of URGE since URGE introduce a further variance introduced by the generation trajectory (AFDPS weights is marginal expectation of URGE weight as we showed in the theory. However, the advantage of URGE does not lie in having a smaller local weight variance. Rather, it stems from providing an exact Radon--Nikodym correction on path space, while avoiding the need to compute derivatives or Laplacians of real-world reward functions.  In the inverse problem example, we show that the performance gap between AFDPS and URGE is small, indicating that their variances are very similar. We apologize for not making this point clear in the paper, and we have already updated the paper to include the corresponding discussion.
>
> [1] Higham, Desmond J. "An Algorithmic Introduction to Numerical Simulation of Stochastic Differential Equations." SIAM Review. 2001.
>
> [2] Talay, Denis, and Luciano, Tubaro. "Expansion of the global error for numerical schemes solving stochastic differential equations." Stochastic Analysis and Applications. 1990.
>
> [3] Shakir, Mohamed, et al. "Monte Carlo Gradient Estimation in Machine Learning." Journal of Machine Learning Research. 2020.
>
> [4] Zhu, Qijie, et al. “Training-Free Adaptation of Diffusion Models via Doob's $h$-Transform”. arXiv preprint arXiv:2602.16198, 2026.

---

> > ### Author Rebuttal · Reviewer_FaxC · 2026-04-02
> >
> > Thank you for your clarification; my questions are fully addressed.
> >
> > I also appreciate the authors' honesty on the larger variance of URGE. From highdsight, it is a consequence of the Rao-Blackwell theorem.

---

### Official Review · Reviewer_NND7 · 2026-03-13

**Soundness:** 4
**Presentation:** 3
**Significance:** 3
**Originality:** 3
**Overall Recommendation:** 5
**Confidence:** 4

**Summary:**

The authors tackle the question of designing efficient inference-time reward guidance for controlled generation with diffusion models. Given a pretrained model for unconditional sampling targeting $p_{\rm data}$, the goal is to steer generated samples, at inference, towards a task-dependent reward $r$. In other words, the goal is to guide the pretrained model to target the tilted distribution $q(x) \propto p_{\mathrm{data}}(x) \exp \{r(x)\}$ with no retraining.

To achieve this objective, the authors introduce URGE, a Sequential Monte Carlo--based guidance procedure that aims to overcome the existing methods' limitations for such task. In particular, previous unbiased methods known as AFDPS and FK-Corrector operate in particle space: they propagate particles with the guided diffusion, they then apply Feynman-Kac correction to reweight and resample particles so that the particle system tracks the desired tilted distribution. It follows that such correction rely on a PDE characterization of the tilted flow, which can be expensive or unavailable for black-box neural rewards, as it involves high order derivatives of the reward function.

On the contrary, URGE  lifts the SMC procedure to path space. The main idea is to consider the path measure of three processes: the reference (unconditional) $P$, the guided proposal process (with guidance augmented drift) $P^G$ and the true tilted process whose terminal marginal is $q(x)$ ,$Q$. URGE simulates particles under $P^{G}$ and assigns to each trajectory a weight proportional to the Radon--Nikodym derivative $\mathrm{d} Q/\mathrm{d} P^{G}$, decomposed as $(\mathrm{d} Q/\mathrm{d} P)\,(\mathrm{d} P/\mathrm{d} P^{G})$, where the second factor is given by a Girsanov change of measure. Crucially such expression is free of high order derivatives of the reward function for the reweighting scheme ($\partial_t r, \nabla r, \Delta r$). The algorithm then performs sequential resampling over trajectories using these weights, yielding a particle approximation of the tilted distribution.

**Compliance With Llm Reviewing Policy:**

Affirmed.

**Key Questions For Authors:**

- The theoretical analysis assumes bounded derivatives of the reward and regularity conditions on the reference drift (for the unconditional model). Could the authors clarify how restrictive the assumptions mentioned in the Appendix are in the context of practical diffusion models and black-box rewards?
- As mentioned earlier, do the authors have any insight on a potential trade off between the step size $\Delta t$ and the particle count $N$ ?

**Limitations:**

yes

**Strengths And Weaknesses:**

The article provides a conceptually elegant approach to reward guidance. The motivation is well justified, especially for black-box rewards or rewards for which gradient evaluation is costly. The path-measure viewpoint is an interesting perspective, and the resulting weight design via Girsanov is tested empirically on several benchmarks against similar reward guided SMC schemes.

The paper is well written and easy to follow. It also makes an effort to position URGE relative to other SMC-based correctors and includes a theoretical equivalence result after marginalization.

The theoretical framework appears a bit restrictive as it assumes bounded derivatives of the reward function. Similarly the uniform Lipschitzianity and linear growth for the drift of the reference process should, in my opinion, be precised in the main text as it has implication on the underlying reference process, and therefore on the unconditionnal generative model considered. I also feel that the necessary conditions to apply Girsanov's theorem should be mentioned and discussed in the main text.

Moreover, unbiasedness should be understood as an asymptotic property. The current theoretical analysis does not track the discretization error nor the Monte Carlo error. Providing non-asymptotic error bounds would significantly strengthen the theoretical part. In particular, I am curious to know whether there is a theoretical or empirical tradeoff between the step size $\Delta t$ and the particle count $N$ that the authors might have explored. This question is also linked to the comment you made "in practice we apply SMC resampling at every step to achieve better performance."

---

> ### Author Rebuttal · Authors · 2026-03-31
>
> We thank the reviewer for the thoughtful comments and for recognizing both the motivation and the technical contribution of the paper. We address the reviewer's comments point by point.
>
> ### Regards regularity conditions on the reward and reference drift
>
> We add a discussion of the regularity conditions in the main text, particularly those on the reward function, which we detail as follows. The regularity conditions on rewards (bounded derivatives) are common in related literature, e.g., Assumption 1 in [1] and Assumption (b) in [2]. It is also a common assumption in the Bayesian inversing problem literature [3] and sequential Monte Carlo literature [4]. The regularity of the reference drift, or equivalently the regularity of the score function, is also commonly seen in the analysis of diffusion models, e.g., [5,6]. These are proof-level regularity assumptions, not URGE’s practical operating requirements. Indeed, the practical significance of URGE is precisely that the correction avoids higher-order reward derivatives. We will also state the Girsanov conditions explicitly in the main text rather than leaving them implicit. Thank you for the suggestion, and we apologize for our oversight.
>
> ### Regards Discritization Error and Choice of Guidance
> Quantitatively, the bias of discretization is of order $O(\Delta t)$, which follows from the standard weak error theory for time discretizations of SDEs with the Euler--Maruyama scheme [7,8]. As the step size decreases, the resampling weights remain closer to one, leading to reduced resampling error. More precisely, when the step size is $O(\Delta t)$, the resampling variance is also of order $O(\Delta t)$, which helps control variance growth over time. Consequently, the component of the variance associated with time discretization remains $O(1)$ and does not accumulate as $\Delta t \to 0$. On the other hand, the dependence on the number of particles $N$ follows the standard Monte Carlo scaling: the variance is $O(N^{-1})$, equivalently the root-mean-square error is $O(N^{-1/2})$ [9]. Therefore, the total mean-square error can be decomposed as $\mathrm{MSE}^2[\text{URGE}] = O(\Delta t) + O(N^{-1})$. This would lead to  selecting the number of particles proportion to the time steps.
>
> We agree that better guidance may further reduce the variance. This is, in fact, another motivation for our analysis of AFDPS.  Our analysis suggests that the main difficulty in AFDPS is controlling the variance introduced by the resampling step. In the special case where the function \(G\) is a solution to a Bellman equation (which can be simulated using Monte Carlo methods [10]), It\^o's lemma implies that the reweighting factor is exactly 1. More generally, the magnitude of the log reweighting factor can be bounded in terms of the Bellman error. This analysis also yields a new scaling law that balances the Monte Carlo cost of simulating Doob’s h-transform, the number of particles, and the number of generation steps.
> Formalizing these observations rigorously would require a substantial amount of additional analysis, likely beyond the scope of the current paper. We therefore leave this direction for future work and will add a discussion of it in the revised version.
>
>
> [1] Chen, Haoxuan, et al. "Solving inverse problems via diffusion-based priors: An approximation-free ensemble sampling approach." arXiv preprint arXiv:2506.03979 (2025).\
> [2] Wu, Luhuan, et al. "Practical and asymptotically exact conditional sampling in diffusion models." Advances in Neural Information Processing Systems 36 (2023): 31372-31403.\
> [3] Stuart, Andrew M. "Inverse problems: a Bayesian perspective." Acta numerica 19 (2010): 451-559.\
> [4] Chopin, Nicolas, and Omiros Papaspiliopoulos. An introduction to sequential Monte Carlo. Vol. 4. Cham, Switzerland: Springer, 2020.\
> [5] Chen, Sitan, et al. "The probability flow ode is provably fast." Advances in Neural Information Processing Systems 36 (2023): 68552-68575.\
> [6] Chen, Sitan, et al. "Sampling is as easy as learning the score: theory for diffusion models with minimal data assumptions." International Conference on Learning Representations. 2023.\
> [7] Higham, Desmond J. "An Algorithmic Introduction to Numerical Simulation of Stochastic Differential Equations." SIAM Review. 2001.\
> [8] Talay, Denis, and Luciano, Tubaro. "Expansion of the global error for numerical schemes solving stochastic differential equations." Stochastic Analysis and Applications. 1990.\
> [9] Shakir, Mohamed, et al. "Monte Carlo Gradient Estimation in Machine Learning." Journal of Machine Learning Research. 2020.\
> [10] Zhu, Qijie, et al. “Training-Free Adaptation of Diffusion Models via Doob's $h$-Transform”. arXiv preprint arXiv:2602.16198, 2026.

---

> > ### Author Rebuttal · Reviewer_NND7 · 2026-04-03
> >
> > Thank you for the detailed responses and clarifications provided during the rebuttal phase. The explanations regarding the theoretical aspects and implementation details satisfactorily addressed my questions and improved the clarity of the contribution.
> >
> > Based on these clarifications, I confirm my positive assessment and maintain my original score.

---

### Decision · Program_Chairs · 2026-04-30

**Decision:**

Accept (regular)

**Comment:**

The reviewers agreed that this is a novel approach to reward-guidance, it provides a valuable perspective on current SMC-based guidance,   it is theoretically grounded, and theoretical analysis is solid. There was some concern around limited experimental validation but not enough to change the overall recommendation from a solid accept.

That said, the reference lists must be fixed for camera-ready as it contains several inconsistencies and some errors. Some examples after a brief skim (these are likely not the only ones):

et al. for no reason (multiple locations):
Sebastian Castillo et al. Adaptive guidance: Training- ´
free acceleration of conditional diffusion models. arXiv
preprint arXiv:2312.12487, 2023.

Hao is incorrect, should be Luhuan:
Hao Wu, Brian Trippe, Christian Naesseth, David Blei, and
John Cunningham. Practical and asymptotically exact
conditional sampling in diffusion models. arXiv preprint
arXiv:2302.09325, 2023a.

Mix of abbreviated first names when most are not:
Alexander J Smola, A Gretton, and K Borgwardt. Maximum mean discrepancy. In 13th international conference,
ICONIP, volume 6, 2006.